# General synthesis and atomic arrangement identification of ordered Bi–Pd intermetallics with tunable electrocatalytic CO$_2$ reduction selectivity

Wenjin Guo[1,2], Guangfang Li[3], Chengbo Bai[1], Qiong Liu[2], Fengxi Chen [1] & Rong Chen [1,4] ✉

Intermetallic compounds (IMCs) with fixed chemical composition and ordered crystallographic arrangement are highly desirable platforms for elucidating the precise correlation between structures and performances in catalysis. However, diffusing a metal atom into a lattice of another metal to form a controllably regular metal occupancy remains a huge challenge. Herein, we develop a general and tractable solvothermal method to synthesize the Bi-Pd IMCs family, including Bi$_2$Pd, BiPd, Bi$_3$Pd$_5$, Bi$_2$Pd$_5$, Bi$_3$Pd$_8$ and BiPd$_3$. By employing electrocatalytic CO$_2$ reduction as a model reaction, we deeply elucidated the interplay between Bi-Pd IMCs and key intermediates. Specific surface atomic arrangements endow Bi-Pd IMCs different relative surface binding affinities and adsorption configuration for *OCHO, *COOH and *H intermediate, thus exhibiting substantially selective generation of formate (Bi$_2$Pd), CO (BiPd$_3$) and H$_2$ (Bi$_2$Pd$_5$). This work provides a comprehensive understanding of the specific structure-performance correlation of IMCs, which serves as a valuable paradigm for precisely modulating catalyst material structures.

Bimetallic nanocrystals are emerging as fascinating materials exhibiting novel properties and capabilities due to the synergistic effects between the two metals and the composition-dependent surface structure and atomic segregation behavior[1,2]. Depending on the mixing patterns, only when the second metal atoms diffuse into the lattice of the first one and form metal–metal bonds with long-range ordered atomic arrangement, intermetallic compounds (IMCs) can be obtained[3,4]. Noticeably, in contrast to the more common phase-separated or solid-solution alloys, the ordered intermetallic compounds tend to exhibit different physical and chemical properties even if they get the same elemental composition and atomic ratios. More

importantly, as unique potential catalysts, intermetallic compounds are more suitable than disordered alloys to explore the structure-activity correlations in catalytic processes due to their special surface properties and well-defined atomic arrangements[5–7]. The ordered surface and bulk configurations have great advantages in studying their adsorption behavior and the electron transfer processes of key intermediates, which will determine the overall reaction activity and selectivity during catalysis. For example, the electrocatalytic CO$_2$ reduction process involves multiple protons coupled electron transfer reactions (PCET), so varieties of reduction products can be obtained according to different electron transfer numbers[8]. In addition, it is

[1]State Key Laboratory of New Textile Materials & Advanced Processing Technologies, Wuhan Textile University, 430200 Wuhan, China. [2]School of Chemistry and Environmental Engineering, Wuhan Institute of Technology, 430205 Wuhan, PR China. [3]Key Laboratory of Material Chemistry for Energy Conversion and Storage (Ministry of Education), Hubei Key Laboratory of Material Chemistry and Service Failure, Huazhong University of Science and Technology, 430074 Wuhan, PR China. [4]Henan Institute of Advanced Technology, Zhengzhou University, 450002 Zhengzhou, PR China. ✉e-mail: rchenhku@hotmail.com

generally believed that the final product strongly depends on the adsorption configuration and energy of the key intermediates on the catalyst surface[9]. Therefore, it is feasible and reasonable to fine-tune the surface properties and arrangement of the catalysts at the atomic level to alter the reaction pathway and obtain the target product. Undoubtedly, IMCs provide a desirable platform for atomic-scale structural design and in-depth understanding of the structure-performance correlations in catalyst materials.

Unfortunately, the preparation of well-defined intermetallics is much more complicated than that of disordered alloys. It is not easy to synchronously control the nucleation and growth of two different metals to form an ordered arrangement due to their different thermodynamic and kinetic properties under the same reaction conditions[10,11]. Not any two metals can be co-reduced to synthesize IMCs. Therefore, it is necessary to choose appropriate metals and reduction systems so that they have more synchronous reduction processes to avoid separate nucleation of the two metals to form core-shell[12,13] or heterogeneous structures[14,15] rather than IMCs. Moreover, the overall diffusion process is strongly affected by metal precursors, reaction temperature, time, pH, etc. Conventional alloy metallurgy methods not only mostly involve high-temperature melting and annealing or dangerous reduction gases, but also often produce disordered alloys with uneven sizes and non-unitary compositions[16,17]. Bi-based bimetallic catalysts are emerging as fascinating materials with remarkable catalytic properties. As perfect catalyst candidates, if a Bi-based intermetallics family with tunable composition and atomic arrangement could be achieved, it is definitely of great significance for a systematic and deeper understanding of the structure-activity correlation for specific catalytic reaction due to their special surface properties and well-defined atomic arrangements. To date, although various IMCs such as Bi-Pb[18], Bi-Mo[19], Bi-Ni[20] have been successfully prepared, a controllable synthesis of a Bi-based intermetallics family with tunable composition and atomic arrangement via a simple and general wet chemical route is barely investigated because it is difficult to achieve the synthesis of multiple IMCs with different structures under the same synthesis system. Furthermore, a comprehensive understanding of the structure-activity correlation in catalytic reactions for different phases within intermetallics families remains limited. Therefore, the development of a simple and general system for the controlled synthesis of ordered IMCs still poses a huge challenge.

In this work, we have developed a simple and convenient solvothermal co-reduction methodology for the preparation of structurally ordered family of Bi−Pd intermetallics by means of massive attempts. A variety of Bi-Pd IMCs including monoclinic and tetragonal $Bi_2Pd$, hexagonal BiPd and $Bi_3Pd_5$, monoclinic $Bi_2Pd_5$, rhombic $Bi_3Pd_8$ and orthorhombic $BiPd_3$ are well-determined by a combination investigation based on double Cs-correctors transmission electron microscopy (Double CS-corrected TEM), XAS and XRD. Moreover, the Bi−Pd IMCs family is a promising candidate for the investigation of catalytic $CO_2$ reduction reactions due to their steerable ability of the $CO_2$ adsorption, electron transfer, geometry and coordination environment. The analysis of the surface atomic structure, adsorption energies and configurations of the key intermediates in the catalytic processes via the theoretical calculations and in-situ techniques has thoroughly unraveled the intrinsic structure-dependent selectivity and structure-performance correlations, providing an enlightening case study in this area.

## Results
### Materials synthesis and characterization

In the synthesis of Bi-Pd IMCs, we have run hundreds of experiments to explore the controllable synthesis strategy of ordered Bi−Pd IMCs with different atomic compositions and arrangements by changing the reaction parameters including reaction time, temperature, Bi/Pd molar ratio, and the type of Pd precursors (Supplementary Figs. 1–6). Six different Bi−Pd ordered intermetallic compounds (IMCs) have been successfully prepared by co-reduction of Bi and Pd salts via a general and facile solvothermal method (S1–S6). The powder X-ray diffraction (PXRD; Fig. 1b) patterns of the six samples all exhibit good crystallinity. No distinct diffraction peak is observed for individual metals, instead new Bragg diffraction peaks appear intuitively, suggesting the formation of alloy compounds rather than phase-separated alloys. Moreover, the spectra from S1 to S6 are in perfect agreement with the standard PXRD card, including $Bi_2Pd$, BiPd, $Bi_3Pd_5$, $Bi_2Pd_5$, $Bi_3Pd_8$ and $BiPd_3$,

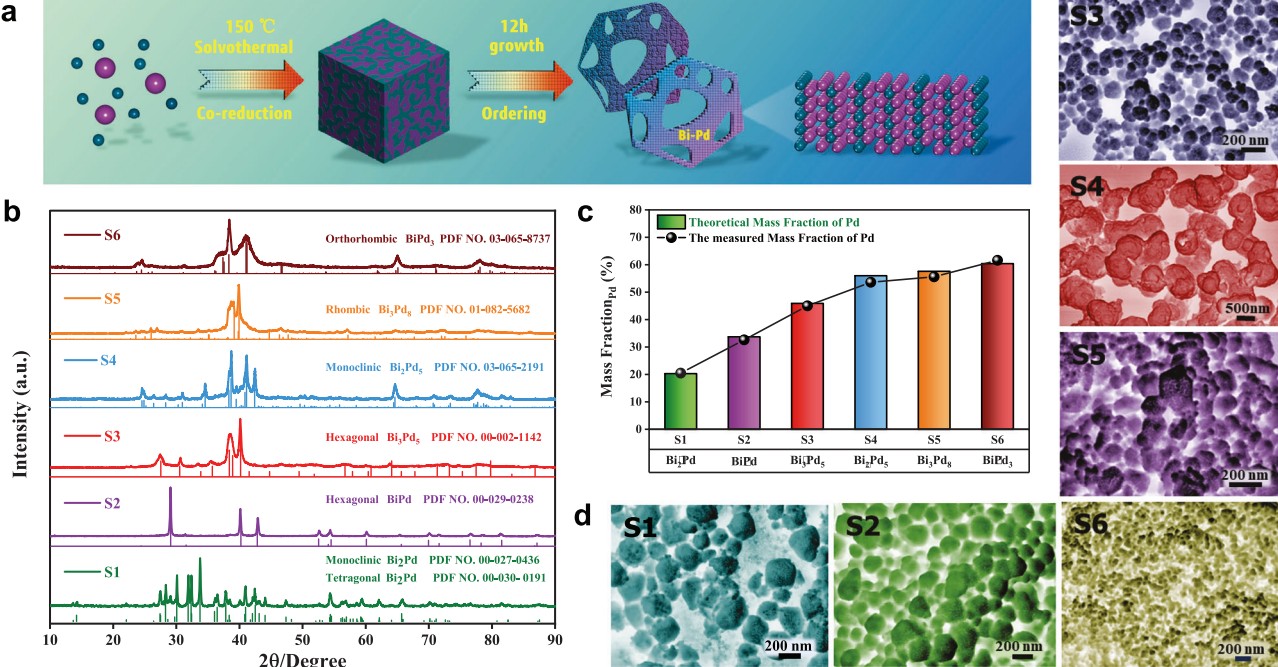

**Fig. 1 | Preparation and structural characterizations of Bi−Pd IMCs. a** Schematic diagram for the synthesis process. **b** XRD patterns. **c** AAS characterization. **d** SEM images.

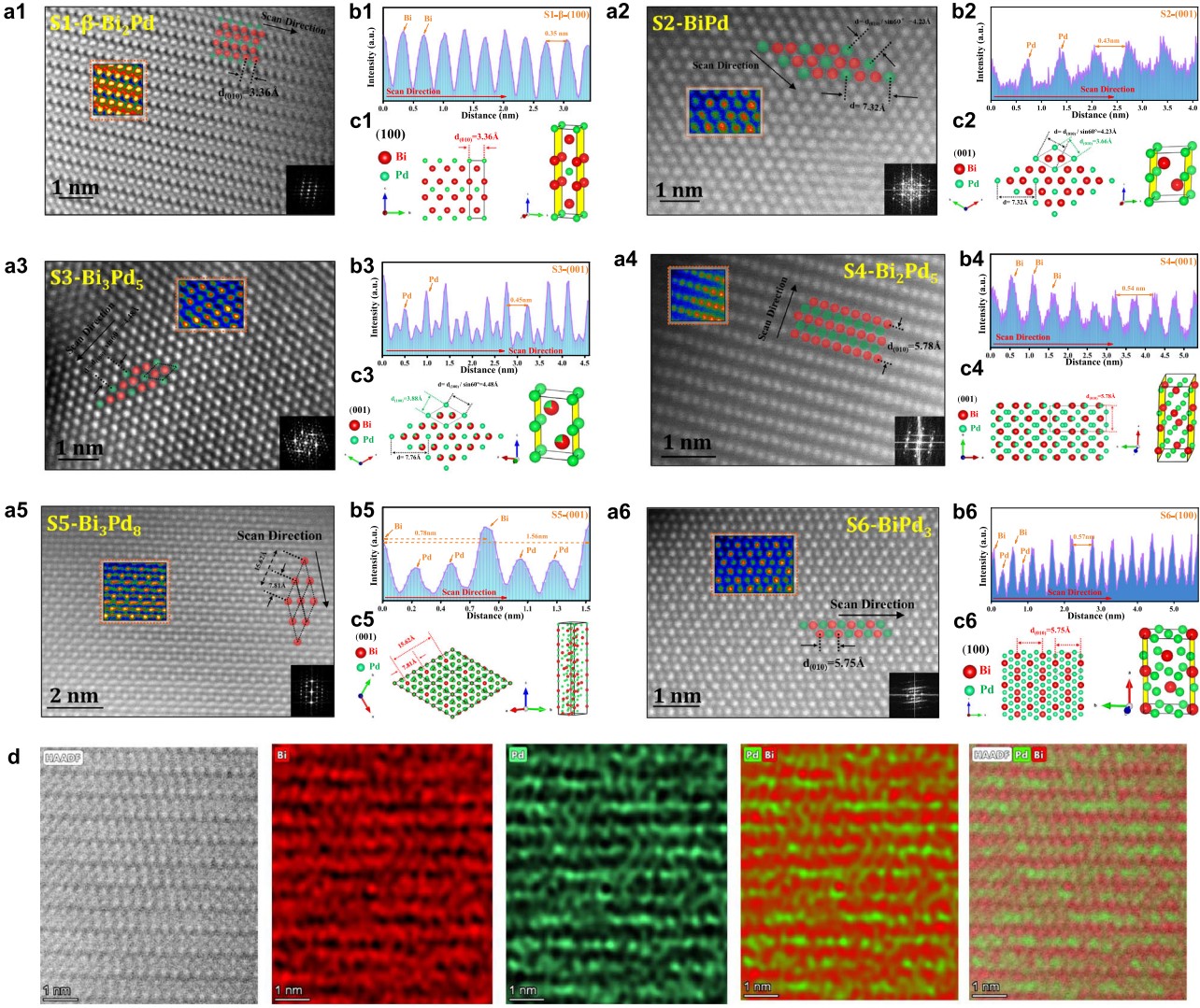

**Fig. 2 | The ordered structure and atomic arrangement of Bi−Pd IMCs. a1−a6** Aberration-corrected HAADF−STEM images of Bi−Pd IMCs (S1 to S6). Inset: corresponding FFT pattern, bright contrast file in the orange box and diagram of atomic arrangement and spacing. **b1−b6** Intensity profiles measured from HAADF−STEM images and (**c1−c6**) corresponding crystal structure (red and green spheres represent Bi and Pd atoms). **d** Atomic resolution EDX mapping of S1-Bi$_2$Pd IMCs.

demonstrating the successful synthesis of Bi−Pd IMCs in the crystal structure. Meanwhile, atomic absorption spectroscopy (AAS; Fig. 1c) quantifies that the elementary compositions of these samples were much the same with the standard chemical formula, which further verifies the accurate synthesis of the IMCs. Figure 1d shows the scanning electron microscope images (SEM) of the six Bi−Pd IMCs, which reveals a uniform size and morphology.

In order to confirm the ordered Bi−Pd IMCs more directly, the double Cs-correctors transmission electron microscopy (Double CS-corrected TEM) is employed to visualize the atomic arrangement of the six samples. Incredibly, as shown in the high-angle annular dark field (HAADF; Fig. 2) image, the whole lattice of all the samples shows highly ordered periodic arrangement of alternating bright and dark atoms. While the larger bright atoms and the smaller dark atoms are assigned to Bi and Pd atoms respectively, according to their vastly different atomic number[21]. Taking Bi$_2$Pd (S1) as an example (Fig. 2a1), combined with the atomic bright contrast files and atomic model overlay in HAADF, it can be found that two rows of bright atoms are followed by a row of dark atoms, and the adjacent rows of dark atoms are separated by one atom in the vertical direction. This special arrangement matches perfectly with the (100) plane of tetragonal Bi$_2$Pd (Fig. 2c1), and can also be validated in the corresponding fast

Fourier-transform (FFT) pattern insert in Fig. 2a1. Moreover, the intensity profile along the scan direction (Fig. 2b1) transformed from the HAADF images presents a periodic oscillation pattern, and the spacing between two adjacent bismuth atoms is measured to be 3.36 Å, which could be assigned to the standard lattice spacing of (010) plane that perpendicular to (100). More immediately, the EDX mapping at atomic resolution (Fig. 2d) also exhibits alternating bright and dark rows of atoms, confirming the (100) plane of Bi$_2$Pd. In the same way, the highly ordered atomic arrangements and exposed crystal planes of the other five Bi−Pd IMCs are finally determined by comparing the bright contrast files, the corresponding FFT patterns in the HAADF images, and the measured characteristic atomic spacing to the corresponding standard crystal spacing (Fig. 2).

To further understand the Bi−Pd IMCs from the electronic structure, X-ray absorption spectra (XAS) of the six samples at Bi L3-edge and Pd K-edge are investigated. As the X-ray absorption near-edge spectra (XANES) shown in Supplementary Fig. 8, the overall electronic structure and spectral trends are similar with the corresponding metals rather than oxides, presenting that the metallic state is the main state of all IMCs[22–24]. However, due to the difference in the electronegativity of Bi and Pd, the absorption edges at the Bi L3-edge and Pd K-edge are shifted to different degrees, indicating the presence of

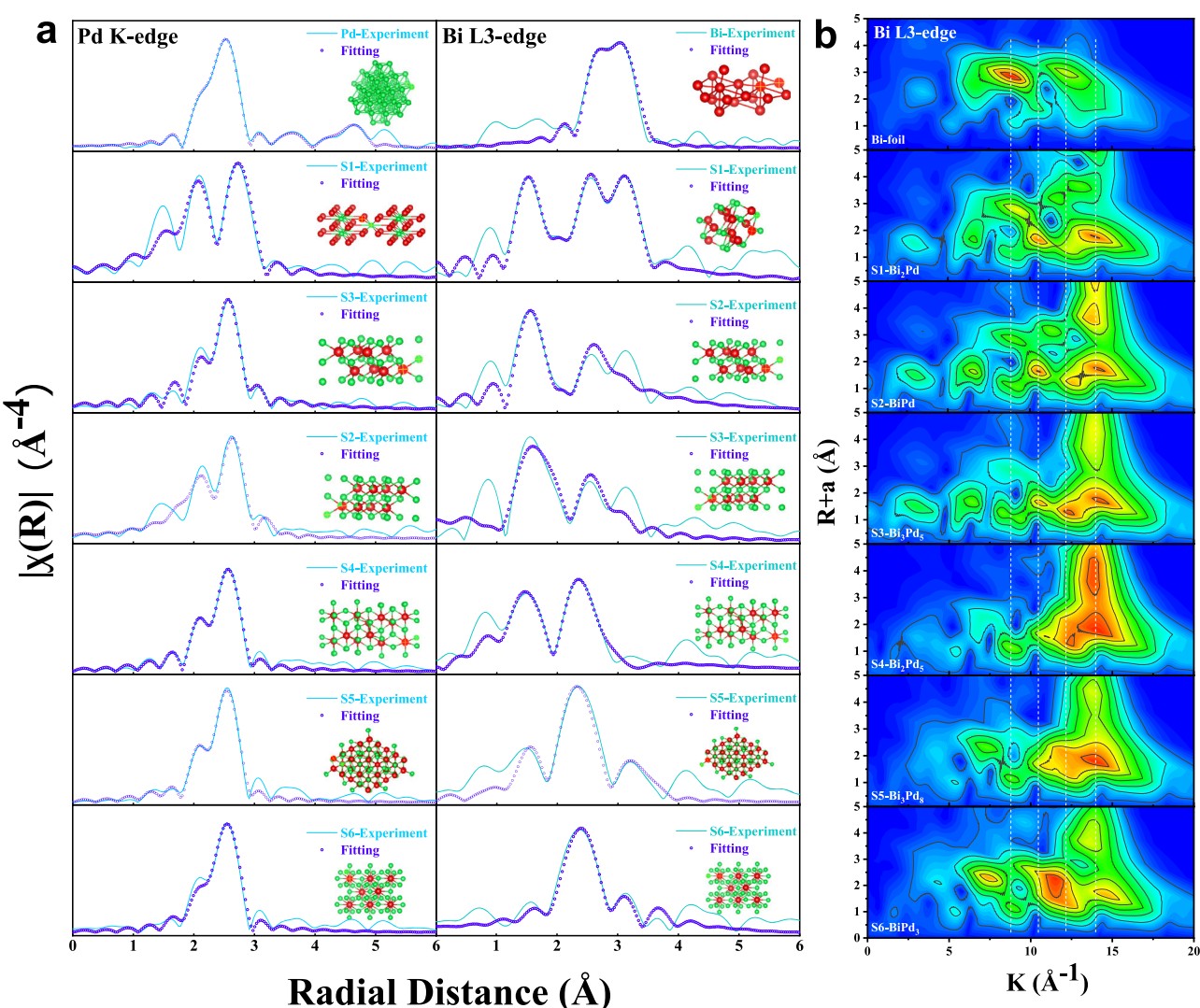

**Fig. 3 | X-ray absorption spectroscopy analysis of Bi–Pd IMCs. a** Fourier-transform EXAFS spectra fittings for Bi–Pd IMCs at Pd K-edge and Bi L3-edge. **b** WT–EXAFS Bi L3-edge spectra for Bi–Pd IMCs and Bi foil reference.

charge transfer between the two elements, reflecting the average valence state of the six IMCs. Besides, K and R spaces of the extended X-ray absorption fine structure (EXAFS) spectra (Supplementary Fig. 8) are further performed to study the bonding states and coordination environments of the six samples. Noticeably, the FT-EXAFS results of IMCs are significantly different from those of single metals, as the prominent peak assigned to the Bi–Pd path at the Bi L3-edge in the first shell of the IMCs is shorter than the Bi-Bi path at 3.0 Å for the Bi foil. More importantly, the structure fitting results of R space and corresponding parameters (Fig. 3a, Supplementary Tables 1, 2) present highly consistence with the standard crystals in bond length and coordination number, which verifies the successful synthesis of ordered IMCs again from the electronic structure. In addition, the analysis of the continuous Morlet wavelet transform for the six samples is shown in Fig. 3b and Supplementary Fig. 9. Clearly, the WT maximum for Bi–Pd IMCs at the Bi L3-edge exhibit lower R-values than that for Bi foils, which corresponds to the formation of shorter Bi–Pd bond. Moreover, for the Bi–Pd path, the increased K-values from S1 to S6 match the higher Bi–Pd coordination numbers among them.

Energy-dispersive X-ray spectroscopy (EDX) mapping and line scan techniques are performed to characterize the element content and distribution. As shown in Supplementary Figs. 10–15, the Bi–Pd atomic fraction determined by STEM-EDX is quite close to the standard

atomic ratio and AAS results. The EDX line scan shows that Bi and Pd are well distributed throughout the nanocrystals, and intensity trough signal in the center of the crystal indicates the hollow structure of IMCs, which might be formed by the Kirkendall effect during the reduction process[25,26]. Moreover, the EDX mapping (Supplementary Figs. 10–15) also confirms the uniform distribution of the two elements, indicating the formation of alloys rather than the phase separated heterostructure or core-shell structure. And the atomic resolution EDX mapping further excludes the formation of random alloys[27,28].

## $CO_2$ electroreduction performance

To evaluate the $CO_2RR$ performance of six different Bi–Pd IMCs dropped on the gas diffusion electrodes (GDE), the electrolytic reactions occur at the triple phase boundary of the flow cell with alkaline media. Supplementary Fig. 16 depicts the linear sweep voltammetry (LSV) of six samples in Ar and $CO_2$ atmosphere. The polarization curves show that all six IMCs in $CO_2$ achieve lower onset-potential and higher current densities at low over-potential than in Ar, reflecting the high reactivity of the Bi–Pd IMCs in $CO_2RR$. While the current density in Ar for S3, S4 and S5 is greater than that in $CO_2$ with the increased over-potential, indicating the enhanced hydrogen evolution reaction (HER). Moreover, by comparing the LSV curves of the six samples in $CO_2$

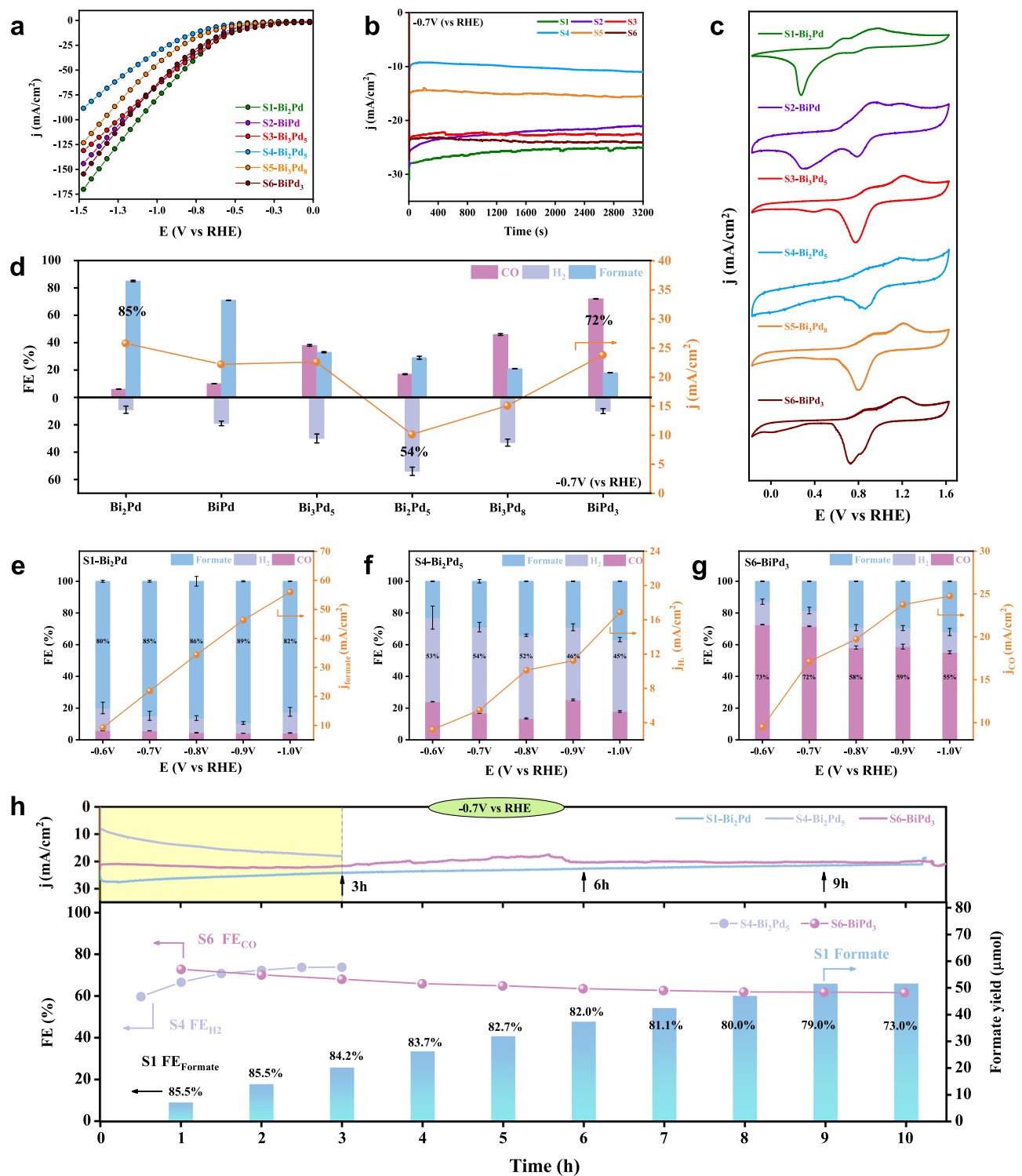

**Fig. 4 | Catalytic performances and stabilities. a** LSV curves (**b**) constant potential (−0.7 V vs. RHE) electrolysis (**c**) CV curves (**d**) FE and current density at −0.7 V (vs. RHE) of the Bi–Pd IMCs. **e** FE and current density of S1 (**f**) S4 (**g**) S6 in all potentials range (**h**) and stability test at −0.7 V vs. RHE in flow cell. Reaction conditions: catalyst mass loading = 1 mg cm⁻², effective area = 1.0 × 1.0 cm², flow rate of $CO_2$ = 30 mL min⁻¹, flow rate of electrolyte = 20 mL min⁻¹, the electrolyte =1 M KOH.

(Fig. 4a), it can be found that the current density distribution from S1 to S6 exhibits a trend of decreasing first and then increasing, and the current density of S1 and S6 is much higher than that of S4, reaching over 150 mA cm⁻² at −1.5 V (versus RHE, the same hereinafter). Figure 4c shows the cyclic voltammetry (CV) curves of six IMCs. In the forward and backward scans, two pairs of typical surface redox waves appear, while no significant hydrogen underpotential deposition is observed.

In addition, the ratio of the two pairs of redox waves varies between the different samples from S1 to S6, indicating the different content and distribution of the two elements on the surface of these IMCs.

The constant potential electrolysis at −0.7 V is further performed to compare the activity of the different Bi−Pd IMCs. As shown in the *i−t* curves (Fig. 4b), all IMCs exhibit high current densities with only slight fluctuations, indicating superior reactivity and chronoamperometry

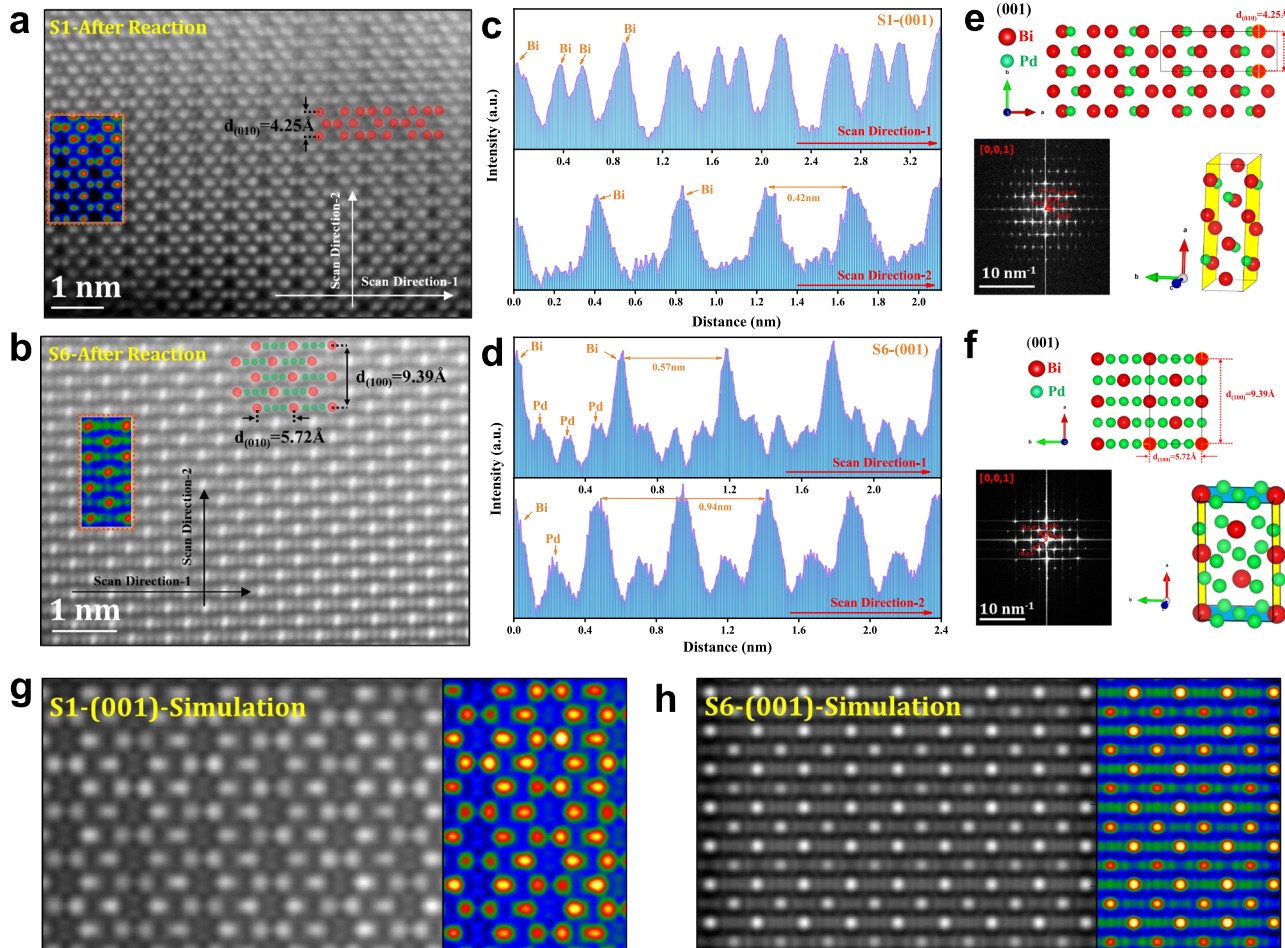

**Fig. 5 | Structural characterization of the catalyst after reaction. a, b** Aberration-corrected HAADF−STEM images of S1 and S6 after reaction. Inset: bright contrast file in the orange box and diagram of atomic arrangement and spacing. **c, d** Intensity profiles measured from HAADF−STEM images and (**e, f**) corresponding FFT pattern and crystal structure (red and green spheres represent Bi and Pd atoms) of S1 and S6 after reaction. **g, h** Simulated HAADF image of S1 and S6 (001) plane.

stability. Besides, the products analysis further indicates that the Bi−Pd IMCs display not only superior reaction activity and stability, but more importantly a structure-dependent selectivity in $CO_2RR$ (Fig. 4d). Among these Bi−Pd IMCs, the S1-$Bi_2Pd$ gets the highest Faradaic efficiency (FE > 85 %) for $CO_2$-to-formate electroreduction, but the FE decreases gradually from S1 to S6. On the contrary, the FE of IMCs for CO increases, and the S6-$BiPd_3$ demonstrates the highest $CO_2$-to-CO selectivity (FE > 70 %). While for the unavoidable HER during electro-catalysis, consistent with the trend in the total current density, it shows a volcano-like distribution among the six IMCs, with S4-$Bi_2Pd_5$ having the highest $H_2$ selectivity compared to S1 and S6 (FE > 50 %). In other words, by tuning the structure and composition of the Bi−Pd IMCs, we achieve the structure-dependent transitions toward CO, formate and $H_2$, respectively. Under the increasing over-potential (Fig. 4e-g), S1 could still maintain a high FE over 80 %, and S4 still has a dominant hydrogen evolution tendency. At the same time, the FE of S6 starts to decay above −0.8 V due to CO poisoning[29]. On the other hand, the products distribution at different voltage windows could be more intuitively verified by comparing the $i−t$ curves and partial current densities under different over-potentials (Supplementary Fig. 17).

As a step further, we investigate the long-term stability of S1, S4 and S6 at −0.7 V (Fig. 4h). Under the constant potential electrolysis for 10 h, S1 and S6 shows no significant loss of activity, while the current density of S4 double in only 3 h, and the FE undulation of $H_2$ exceeds 20%. In contrast, the corresponding FE decays for S1 and S6 are less than 10 %, and the formation rate of formate is essentially stable within

10 h, indicating that S1 and S6 are better electrochemical stable than S4. On the other hand, the utilization of anion exchange membranes is inevitably accompanied by formate crossover or carbon loss during the $CO_2RR$, leading to a decrease in device efficiency. Therefore, we implemented several strategies, including regular electrolyte refreshing, membrane replacement, and carbon paper retreatment to mitigate these significant limitations. As depicted in Supplementary Fig. 18a, at −100 mA cm⁻² electrolysis, through periodic treatments, the S1 sample exhibited superior stability for nearly 60 hours with small changes in formate crossover and carbon loss, and a well-maintained formate selectivity (FE ~ 70 %). However, prolonged usage leads to gradual hydrophobicity loss (Supplementary Fig. 18b) of the gas diffusion layer, that hinders $CO_2$ diffusion to reaction sites while promoting hydrogen evolution reaction and the electrode flooding at the reactive three-phase interface, thus limiting the further stability testing within the current flow cell setup. Moreover, the restructuring phenomenon usually occurs during electrochemical $CO_2RR$ of metal catalysts which have been widely studied in literatures[30,31]. In this work, the XRD patterns of the IMCs before and after long-term stability test manifest no recognizable structure change (Supplementary Fig. 19). We further employ HAADF-STEM to study the surface structure of spent S1 and S6 from an atomic level. As shown in Fig. 5, both S1 and S6 after reaction keep highly ordered periodic arrangements with alternating bright and dark atoms. By analyzing the special atomic arrangement and the bright contrast files (Fig. 5a, b), the ordered arrangements are well matched with the (100) plane of S1 and S6. The

further comparison of intensity profiles (Fig. 5c, d) along different directions and corresponding fast Fourier-transform (FFT) patterns (Fig. 5e, f) are in excellent agreement with the standard crystal spacing. Finally, we simulate the arrangement of (100) plane of S1 and S6, respectively, which exhibits exactly the same pattern as the HAADF-STEM images we took (Fig. 5g, h). All the results indicate that the Bi–Pd IMCs are extremely stable without compositional or structural variations during the long-time electrolysis.

## Mechanism investigations

It is well known that particle size and morphology might have great influences on catalytic performance. We thus synthesized a series of different-sized Bi–Pd IMCs to carry out $CO_2RR$ to rule out size effect. All samples display irregular grain-like morphologies, while the crystal size of $Bi_2Pd$ (S1), $Bi_2Pd_5$ (S4) and $BiPd_3$ (S6) IMCs can be fine-tuned by varying the content of surfactant PVP. However, we do not observe significant changes in both activity and selectivity of Bi–Pd IMCs with same composition but different sizes for electrocatalytic $CO_2RR$ (Supplementary Figs. 20–22). Those results indicate the particle size has a negligible effect on $CO_2RR$ performance. To gain deeper insight into the structure-dependent selectivity of different IMCs, we perform density functional theory (DFT) calculations to understand the reaction processes in $CO_2RR$. As shown in Supplementary Tables 3–5 and Supplementary Figs 23–25, considering the characteristic of the flow cell and alkaline electrolyte, three typical two-electron transfer competing reactions, including $CO_2$-to-formate reaction, $CO_2$-to-CO reaction, HER, are divided into several elementary reaction steps. Then the adsorption energies of the intermediates or transition states in every step on the (100) plane of S1, (001) plane of S4, and (100) plane of S6 are calculated.

For HER, throughout the whole reaction process, the Gibbs free energy diagrams (Supplementary Fig. 26) reveal that the first and the second electron transfer steps are endothermic, while the remaining neutralization and desorption steps are exothermic. In comparison, the first electron transfer step, where the first hydrogen atom is attached to the catalyst surface, is the rate determination step for HER on the three IMCs and has a higher energy barrier than the second step. Moreover, the adsorption energies of the transition states formed during the first electron transfer are compared in all three samples. It is found that S4 gets the lowest activation energy about 1.13 eV compared to S1 (1.66 eV) and S6 (1.23 eV), which is more conducive to HER. Simultaneously, models for the adsorption of these intermediates are shown in Supplementary Fig. 26b, where the adsorption conditions and configurations of these intermediates are significantly different for the three samples. Then we further analyze the adsorption models of the key intermediates, $*H + H_3O_2^-$, along the direction of S1 (100), S4 (001) and S6 (100) (Fig. 6b–d). In particular, the three distinct adsorption modes are clearly observed in the top and side views, where hydrogen atoms are absorbed on the surface of S1, S4 and S6 in the bridge, quadrilateral and triangular modes, respectively. No doubt, the quadrilateral mode adsorption on S4 (001) plane is the most stable state, which evidences the HER tendency on S4.

For $CO_2RR$, the Gibbs free energy diagrams (Supplementary Figs. 27, 28) indicate that after the adsorption of $CO_2$ on the IMCs surface, the next hydrogenation to form *OCHO and *COOH are rate-determining steps for the formation of formate and CO, possessing the highest energy barrier. Different adsorption configurations change the subsequent reaction paths for disparate products. Therefore, the adsorption behaviors of *OCHO and *COOH, two key intermediates, are our focus in $CO_2RR$. For *OCHO, a typical oxyphilic bidentate adsorption configuration, is recognized as an important intermediate for formate production. While *COOH, a carbophilic unidentate adsorption configuration, is deemed to be the key intermediate for CO formation. Among these IMCs, S1 has the lowest *OCHO activation energy (about 1.45 eV compared with 3.57 eV and 2.09 eV for S4 and

S6), and S6 has the lowest *COOH activation energy (about 1.16 eV compared with 1.73 eV and 1.20 eV for S1 and S4), illustrating the different adsorption and hydrogenation preference on the surface of S1 and S6. Meanwhile, in the three views of the bidentate *OCHO adsorption models (Supplementary Fig. 29a–c), we could find that on the S1 (100) plane, the two oxygen atoms are bridged with the two adjacent palladium atoms, thus having shorter C–O bonds (1.266 Å for S1 and 1.269 Å for S4 and S6) and larger angle between C–O bonds (129.9°, 129.3°, 128.8° for S1, S4 and S6), which facilitates the bidentate *OCHO adsorption on $Bi_2Pd$ more stable. However, things are different on S6 (100) plane that the unidentate *COOH adsorption turns to be the dominant intermediate. As shown in Supplementary Fig. 29d–f, since the S6 (100) gets a planar structure, the Pd atoms on the surface are saturated coordinated, which is more conductive to adsorption with carbon atom in a unidentate pattern rather than bridging with two oxygen atoms to form *OCHO. Moreover, according to the projected partial density of states (PDOS) analysis near the Fermi level (Supplementary Fig. 30), the d-band center of S6 is at −1.92 eV while that of S4 and S1 shifts away from the Fermi level at −2.05 and −2.29 eV, confirming the weakened binding strength of materials to intermediates. Whereas, the weaker the binding, the more inclined to form *OCHO in bidentate form. Comparatively, it would facilitate the formation of *COOH in unidentate form when the d-band center moves upward, and enhanced the CO production on S6 (100) plane.

Focusing on each reaction pathway, we have compared the free energy longitudinally among the three IMCs, and have found that the different surface arrangements of Bi and Pd atoms would directly affect the adsorption configuration of the intermediate, resulting in *H being more stable on S4, *OCHO on S1, and *COOH on S6. In actual electrocatalysis, however, the three paths occur simultaneously and compete each other. Therefore, in order to more precisely verify the structure-dependent selectivity, the free energies of the three paths should also be compared horizontally for each sample. As shown in Fig. 6a, for S1, the rate-determining step of the formate pathway has a lower energy barrier than the other two competing reactions (1.59 eV for formate, 1.73 eV and 1.66 eV for CO and $H_2$), confirming that the formate path on S1 (100) plane is thermodynamically dominant. Similarly, for S4, the HER has a significantly more favorable reaction propensity than $CO_2RR$, with a rate-determining step energy barrier of only 1.13 eV, much lower than 3.57 eV for the formate path and 1.20 eV for the CO path. For S6, the energy barrier of the rate-determining step is about 1.14 eV for the CO path, which is lower than 2.09 eV for the formate path and 1.23 eV for HER. These results are in good agreement with the structure-dependent selectivity over the three IMCs.

To further verify these key intermediates and reaction pathways, in situ Fourier-transform infrared spectroscopy (FTIR) is further performed. As shown in Fig. 6e and Supplementary Fig. 31, with the increased potentials, characteristic peaks located at 1574 and 1098 $cm^{-1}$ are observed in the S6 sample, which is attributed to the vibrations of *COOH and *CO[29,32,33], but not appears in the S4 sample, confirming the proposed $*CO_2 \rightarrow *COOH \rightarrow *CO \rightarrow CO$ reaction pathway in S6. Likewise, the characteristic peak of *OCHO at 1467 $cm^{-1}$ appears in S1 from -0.65 V[34,35], which also verifies that S1 possesses high selectivity for the formate conversion through the *OCHO pathway. This evidence strongly supports the reaction path and mechanism in the theoretical calculation.

## Discussion

In summary, we have presented a general reduction system tailored to the atomic arrangement and compositional stoichiometry of up to six different IMCs, including $Bi_2Pd$, $BiPd$, $Bi_3Pd_5$, $Bi_2Pd_5$, $Bi_3Pd_8$ and $BiPd_3$. The ordered structure and atomic arrangement of different Bi–Pd IMCs have been meticulously confirmed by XRD, XAS and double CS-corrected TEM. Using electrocatalytic $CO_2RR$ as a model reaction, different IMCs in the Bi–Pd family enable a structure-dependent

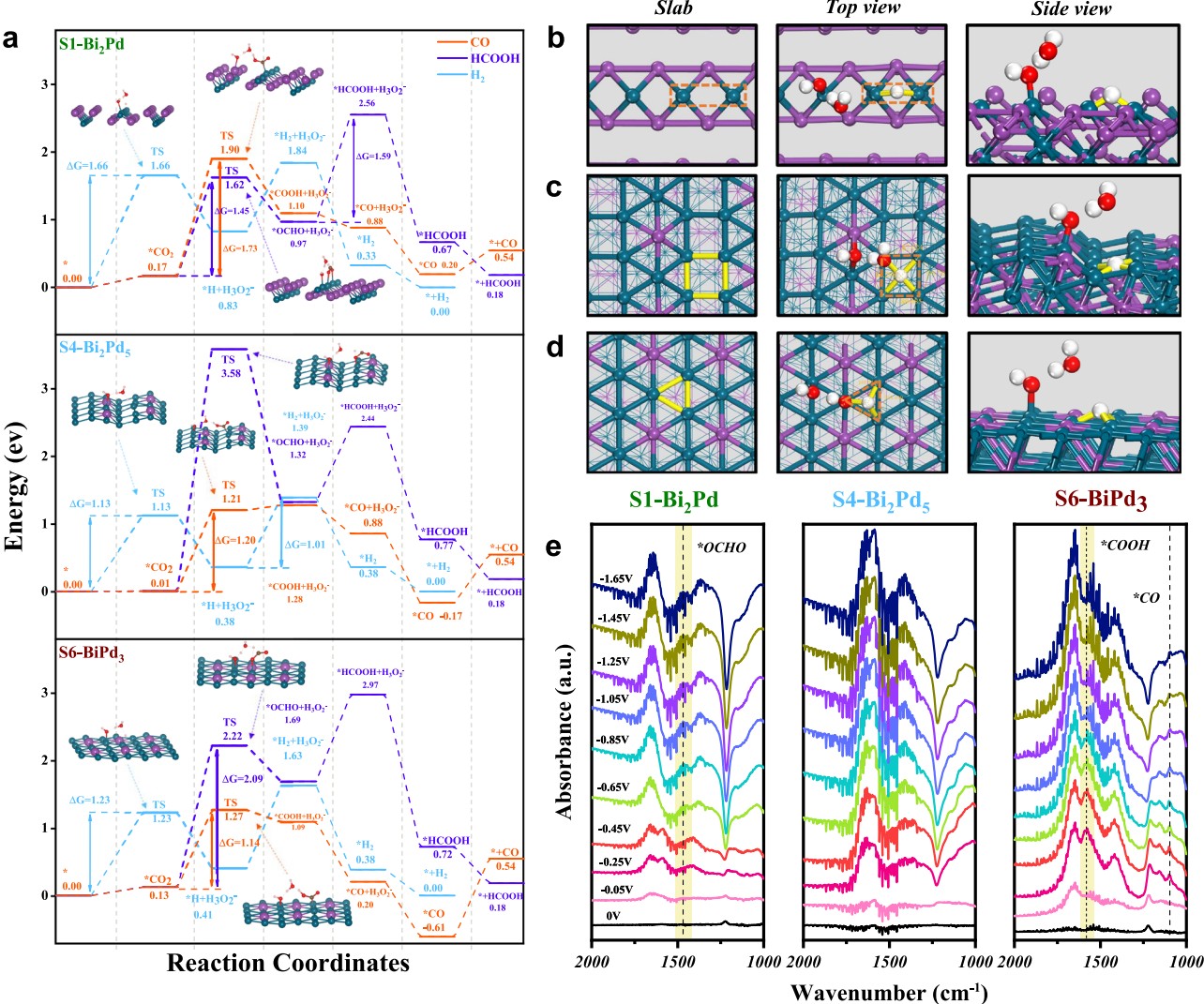

**Fig. 6 | The Reaction mechanism. a** DFT-calculated free energy diagrams for S1, S4 and S6. **b** The optimized atomic structure models of adsorbed *H showed along different directions on S1(100) surface, (**c**) S4(001) surface and (**d**) S6(100) surface. (Red and white balls inside represent O and H atoms). **e** Potential-resolved in situ FT-IR spectra for S1, S4 and S6.

selectivity for $CO_2$-to-formate, $CO_2$-to-CO reactions and HER in flow cells. Theoretical calculations and in-situ FTIR further demonstrate that different arrangement of bismuth and palladium atoms on the surface stabilize the adsorption of bidentate *OCHO on S1, adsorption of unidentate *COOH on S6, and quadrilateral form adsorption of *H on S4, revealing the great impact of atomic-scale regulation for ordered IMCs to the intermediate adsorption. Overall, this work is the first example for controllable synthesis of six different ordered Bi−Pd IMCs and being studied as a platform to understand the structure-activity relationship, which is highly desirable for catalysis studies.

## Methods
### Synthesis of Bi−Pd intermetallics
To prepare $Bi_2Pd$, $Bi(NO_3)_3 \cdot 5H_2O$ (72.8 mg, 0.15 mmol) was first dissolved in $HNO_3$ solution (2 mL, 1 M), then PVP (60 mg, Mw=1000) and EG (10 mL) were injected into this solution, followed by a vigorous stirring for 30 min. subsequently, $Pd(NO_3)_2 \cdot 2H_2O$ (20 mg, 0.075 mmol) was added into the mixed solution. After stirring for another 30 min, the solution was transferred into a Teflon-lined stainless-steel autoclave to perform solvothermal process at 150 °C for 12 h. At the end of reaction, the obtained sample was collected by centrifugation, washing with ethanol and finally being dried at 40 °C for characterization.

For the other IMCs, including $BiPd$, $Bi_3Pd_5$, $Bi_2Pd_5$, $Bi_3Pd_8$ and $BiPd_3$, the synthesis conditions are roughly the same with some slight variations, and the specific parameters are given in the table below. The obtained six kinds of Bi−Pd IMCs are dubbed simply as S1 to S6. (Table 1)

### Preparation of working electrodes
10 mg catalyst was dispersed in 1 mL of ethanol (containing 50 μL of 0.25 wt% Nafion solution) and sonicated to form a uniform ink, which was slowly dropped onto a $1 \times 3\,cm^2$ gas-diffusion layer (GDL, SIGRA-CET 29BC grade) then dried under room temperature to achieve $1\,mg\,cm^{-2}$ loading of catalyst to form the gas-diffusion electrodes. The following electrochemical experiments were carried out on an electrochemical workstation with three-electrode (Gamry Reference 3000, USA).

### $CO_2$ electroreduction testing in flow cells
The CO2RR electrocatalysis was performed in a flow-cell consisting of GDE cathode, platinum plate anode, Ag/AgCl reference electrode, 1 M KOH electrolyte and an anion exchange membrane (FAB-PK-130, Fumatech). And the effective area of both anode and cathode compartment were $1.0 \times 1.0\,cm^2$. The flow rate of the $CO_2$ at the gas chamber was $30\,mL\,min^{-1}$ and flow rate of the electrolyte was

**Table 1 | Synthesis parameters of six kinds of Bi–Pd IMCs**

| Sample | Precursor type | Molar ratio of Bi/Pd | Nitric acid |
|---|---|---|---|
| S1-Bi$_2$Pd | Pd(NO$_3$)$_2$.2H$_2$O | 2:1 | 2 mL |
| S2-BiPd | Pd(OAc)$_2$ | 1:1 | 4 mL |
| S3-Bi$_3$Pd$_5$ | Pd(OAc)$_2$ | 3:5 | 2 mL |
| S4-Bi$_2$Pd$_5$ | Na$_2$PdCl$_4$ | 2:5 | 4 mL |
| S5-Bi$_3$Pd$_8$ | C$_{10}$H$_{14}$O$_4$Pd | 3:8 | 2 mL |
| S6-BiPd$_3$ | Pd(NO$_3$)$_2$.2H$_2$O | 1:3 | 2 mL |

20 mL min$^{-1}$. Electrode potentials were rescaled to the RHE reference by:

$$E(vs\ RHE) = E(vs\ Ag/AgCl) + 0.197V + 0.0591 \times pH$$

The gaseous products were analyzed by a gas chromatography (GC9720Plus, Fuli Instruments) and the liquid product were analyzed by ion chromatography (940 Professional IC Vario, Metrohm) that the anion column was Metrosep A Supp7-250/4.0, the leachate was 36 mmol/L Na$_2$CO$_3$, and the flow rate was 0.8 mL/min.

### XAFS measurements

The XAFS measurements were carried out at Bi L3-edge and Pd K-edge on the BL14W1 beamlines and BL11B beamlines with Si (111) crystal monochromators at the Shanghai Synchrotron Radiation Facility (SSRF) (Shanghai, China). Before the analysis at the beamline, samples were pressed into thin sheets with 1 cm in diameter and sealed using Kapton tape film. The XAFS spectra were recorded at room temperature using a 4-channel Silicon Drift Detector (SDD) Bruker 5040. Negligible changes in the line-shape and peak position of the XANES spectra were observed between two scans taken for a specific sample. The EXAFS spectra were recorded in transmission mode. The data extraction and fitting were carried out by using Demeter program[36]. As for XANES, the experimental absorption coefficients as a function of energies μ(E) were processed by background subtraction and normalization procedures. As for k$^3$-weighted EXAFS, the first-shell approximation was adopted to analyze the Fourier transformed data in R space by taking the parameters of coordinated number (N), bond length (R, A), Debye–Waller factor (σ2, Å$^2$), passive electron factors (S$_0$$^2$) and shift in the edge energy (ΔE$_0$, eV) into consideration.

### Computational methods

The DFT calculations were carried out in the Vienna ab initio simulation (VASP5.4.4) code[37]. The exchange-correlation is simulated with PBE functional and the ion-electron interactions were described by the PAW method[38,39]. The vdWs interaction was included by using empirical DFT-D3 method[40]. The Monkhorst–Pack-grid-mesh-based Brillouin zone k-points are set as $2 \times 2 \times 1$ for periodic cubic structure and $3 \times 3 \times 1$ for periodic hexagonal structure with the cutoff energy of 450 eV. The convergence criteria are set as 0.01 eV A$^{-1}$ and 10$^{-5}$ eV in force and energy, respectively. A 20 Å vacuum layer along the Z direction is employed to avoid interlayer interference. The bottom two layers of atoms are fixed to maintain fixed properties, and the top two layers of atoms are released to simulate the surface structure. The free energy calculation of species adsorption (ΔG) is based on the following equation:

$$\Delta G = \Delta E + \Delta E_{ZPE} - T\Delta S.$$

Herein Δ$E$, Δ$E_{ZPE}$, and Δ$S$ respectively represent the changes of electronic energy, zero-point energy, and entropy that caused by the adsorption of intermediate. The entropy of H$^+$+e$^-$ pair is approximately regarded as half of H$_2$ entropy in standard condition[41].

## Data availability

The source data generated in this study are provided in the Source Data file. Source data are provided with this paper.

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

## Acknowledgements

This work was supported by the National Natural Science Foundation of China (22076149, 92161110) (R.C.), Innovative Team Program of Natural Science Foundation of Hubei Province (2023AFA027) (R.C.) and Special Project of State Key Laboratory of New Textile Materials & Advanced Processing Technologies from Wuhan Science and Technology Bureau (2022013988065204) (R.C.).

## Author contributions

R.C. and W.G. designed the project. W.G. performed all the experiments under the supervision of R.C. and F.C., Q.L., and C.B. did part of characterization and data analysis. G.L. performed all the calculations. All authors discussed the results and contributed to writing the manuscript.

## Competing interests

The authors declare no competing interests.
