## [Peer Review File · Nature Communications]

REVIEWER COMMENTS

Reviewer #1 (Remarks to the Author):

The authors developed a solvothermal method to tune the atomic-level surface structure and the compositional stoichiometry of Bi-Pd IMCs, which were well characterized. The interplay between Bi-Pd intermetallics and the key intermediates of electrocatalytic CO₂ reduction has been investigated by the density functional theory calculations and in-situ Fourier-transform infrared spectroscopy. Below are my specific comments for major revision. After the authors could provide all necessary studies to support their conclusions, we might further consider the publication.

1. Six Bi-Pd IMCs samples shown in Figure 1d have large differences in particle size and morphology, which could significantly affect their CO₂RR performance rather than surface atomic arrangements. Please verify this by additional experiments.
2. The authors need to explain why they performed the constant potential electrolysis at -0.7V while evaluated the stability at -1 V?
3. In Figure 4b, the current densities of samples S4, S5 and S6 increase after a long reaction time, while current densities of other samples decrease with the increase in reaction time. Authors are encouraged to explain this.
4. The authors should give infrared data within a wider wavenumber range, and assign other peaks in the infrared spectrum that vary with the potential, such as the variation of peaks around 1240 cm⁻¹ in the infrared spectrum of S6 as shown in Figure 5e.
5. The authors used anion exchange membranes to test the performance of CO₂RR, so it is necessary to explain how to avoid the carbon loss caused by the crossover of formate and bicarbonate during the long term test. A closed carbon mass balance accounting for all carbon should be provided to assure that the results are valid (determination of gaseous CO₂ at the cathode and anode outlet, of organic molecules in the gas phase in the catholyte as well as anolyte, and of carbonate in the catholyte and anolyte necessary).

Reviewer #2 (Remarks to the Author):

In this paper, the authors reported preparation of a set of BiPd intermetallic catalysts by solvothermal method, by selecting the amount or the type of Pd salt precursors. The author performed detailed

examination for the as prepared catalysts by Aberration corrected TEM and XAFS and DFT calculation and in-situ FTIR were also performed for the catalysts, which is really impressive. There are several reports about the synthesis and application of BiPd intermetallic, which are equally prepared by hydrothermal method, but using different solvent and metal precursors. The concept of the paper, nor the findings seem very new. Thus, the authors may need to highlight the novelty and broad impact of this work.

1. The reports about BiPd intermetallics are shown below:

([https://doi.org/10.1016/S1872-2067\(21\)63999-2](https://doi.org/10.1016/S1872-2067(21)63999-2);

<https://doi.org/10.1002/anie.202109288>; <https://doi.org/10.31635/ccschem.022.202202357>)

2. Usually, restructuring of the metal catalysts occurs during electrochemical CO₂RR occurs. Thus, the real active site may not be BiPd intermetallics at all. Is that possible that the authors perform in-situ examination of XAS, XRD or perform such characterization for the spent samples.

3. Currently, the production of formate is already not very exciting for electrochemical CO₂RR. The selectivity and current density for C₂ products productions can be up to 80% and 1-2 A cm⁻².

4. "However, it remains a huge challenge in the controllable synthesis of IMCs, as diffusing a metal atom into a lattice of another metal to form a regular metal occupancy is no picnic. The formation mechanism for the intermetallic is not well supported by data. Can the authors monitor the formation dynamics for such intermetallics? Why do you have to change the type of Pd precursors to regulate the ratio of Pd to Bi in the intermetallics, but just using only one single Pd precursors.

5. The clarity of the figures provides is quite low and need to be improved.

Reviewer #3 (Remarks to the Author):

The DFT calculation section of the article provides detailed research on HER and CO₂RR reactions occurring on S1, S4, and S6, but there are still several issues:

1. The content of Supplementary Fig. S18 at line 236 does not correspond to the HER reaction described in the text

2. At lines 246 and 247, it is stated that "the next hydrogenation to form * OCHO and * COOH are rate determining steps for the formation of form and CO, possessing the highest energy barrier". However, the

corresponding Supplementary Figures S17 and S18 only indicate the transition state energy barrier but do not indicate the thermodynamic energy barrier for this step of the reaction. Moreover, in Figure S17a, it appears that the energy barrier for * OCHO to * HCOOH is actually higher, which is inconsistent with the textual description; And there is a syntax error in “While the different absorption configurations change the subsequent reaction paths for separate products.”

3. At line 441, Did you increase kpoints when calculating DOS ? Suggest explanation for this.

Dear reviewers,

We would like to thank you for the prompt and thorough review of our manuscript. These comments are valuable and greatly help in the improvement of the manuscript. We have carefully studied the comments, supplemented the experiments and provided a substantive discussion of the results. Please find our point-by-point responses to the reviewer's comments below and the corresponding revisions in *red* text in the revised manuscript.

For reviewer #1:

The authors developed a solvothermal method to tune the atomic-level surface structure and the compositional stoichiometry of Bi-Pd IMCs, which were well characterized. The interplay between Bi-Pd intermetallics and the key intermediates of electrocatalytic CO₂ reduction has been investigated by the density functional theory calculations and in-situ Fourier-transform infrared spectroscopy. Below are my specific comments for major revision. After the authors could provide all necessary studies to support their conclusions, we might further consider the publication.

Comments 1) Six Bi-Pd IMCs samples shown in Figure 1d have large differences in particle size and morphology, which could significantly affect their CO₂RR performance rather than surface atomic arrangements. Please verify this by additional experiments.

Our response: *We thank for the reviewer's extremely insightful comments. It is well known that particle size and morphology might have great influences on catalytic performance. In this work, all the Bi-Pd IMCs display irregular grain-like morphologies, and the particle size of most Bi-Pd IMCs is in the range of 75-100 nm, while obvious size difference exists among S1, S4 and S6 samples. According to the reviewer's suggestion, we synthesized a series of different sized Bi-Pd IMCs to study the size effect on electrocatalytic behavior.*

The particle sizes of Bi₂Pd (S1), Bi₂Pd₅ (S4) and BiPd₃ (S6) IMCs were regulated by varying the content of surfactant PVP during synthesis process, and the experimental details have been added to the Supplementary Information in the revised version. As shown in following Figure R1a-R3a, the XRD diffraction peaks of different samples are in good agreement with the standard patterns of Bi₂Pd, Bi₂Pd₅ and BiPd₃ IMCs, respectively. Figures R1b-R3b show the SEM images and the corresponding size distribution of each IMCs with the same stoichiometry

but different sizes. For Bi₂Pd, the particle size varies from 100 to 200 nm, while a slight current density fluctuation and an almost unchanged formate selectivity were observed toward electrocatalytic CO₂RR (Figure R1c~d). When we switch to the Bi₂Pd₅ sample with relatively large crystal sizes, as shown in Figure R2c~d, different sized Bi₂Pd₅ samples demonstrate negligible activity variations from two perspectives of H₂ FE and constant potential (-0.7 V vs. RHE) electrolysis, even their size difference is hundreds of nanometers. Interestingly, the similar phenomenon also take place on the BiPd₃ samples with relatively small sizes (Figure R3c~d). These results indicate that the particle size has a negligible effect on CO₂RR performance.

In addition, we also carefully compare the activity of Bi₂Pd and BiPd₃ IMCs with similar particle size (Figure R4), which reveals dramatically different activity and selectivity. The Bi₂Pd and BiPd₃ performs catalytic selectivity toward formate and CO, respectively. Besides, Bi₂Pd shows a larger current density than BiPd₃. These results also prove that the surface atomic arrangement rather than the particle size of Bi-Pd IMCs works as the decisive factor of the activity and selectivity of Bi-Pd IMCs. Besides, it has been well demonstrated the activity and selectivity heavily depended on adsorption ability of reaction intermediates of electrocatalyst. Our density functional theory (DFT) calculations show that the surface atomic arrangements endow Bi-Pd IMCs different relative surface binding affinities and adsorption configuration for *OCHO, *COOH and *H intermediate, thus affecting the activity and selectivity. The compelling data further provides another strong evidence that atomic arrangement plays a pivotal role toward CO₂RR performance. Therefore, in this work, we focus more on the synthesis of ordered Bi-Pd IMCs with various atomic compositions and arrangements via a general co-reduction strategy and the structure-activity relation of CO₂RR reactivity and selectivity. We thank the reviewer for this excellent point, and we have added the relevant discussions in the revised manuscript as follows.

“It is well known that particle size and morphology might have great influences on catalytic performance. We thus synthesized a series of different sized Bi-Pd IMCs to carry out CO₂RR to rule out size effect. All samples display irregular grain-like morphologies, while the crystal size of Bi₂Pd (S1), Bi₂Pd₅ (S4) and BiPd₃ (S6) IMCs can be fine-tuned by varying the content of surfactant PVP. However, we do not observe significant changes in both activity and

selectivity of Bi-Pd IMCs with same composition but different sizes for electrocatalytic CO₂RR (Supplementary Figs. S19-S20). Those results indicate the particle size has a negligible effect on CO₂RR performance.” (highlighted with red color in the revised version)

Figure R1 | **a**, XRD patterns **b**, SEM images **c**, FE and current density at -0.7V (vs. RHE) **d**, constant potential (-0.7V vs. RHE) electrolysis of S1 with different sizes.

Figure R2 | **a**, XRD patterns **b**, SEM images **c**, FE and current density at -0.7V (vs. RHE) **d**, constant potential (-0.7V vs. RHE) electrolysis of S4 with different sizes.

Figure R3 | **a**, XRD patterns **b**, SEM images **c**, FE and current density at -0.7V (vs. RHE) **d**, constant potential (-0.7V vs. RHE) electrolysis of S6 with different sizes.

Figure R4 | FE and current density at -0.7V (vs. RHE) of S1-5 and S6-1.

Comments 2) The authors need to explain why they performed the constant potential electrolysis at -0.7V while evaluated the stability at -1 V?

Our response: *We appreciate for the reviewer's preciseness. In fact, we indeed studied constant potential electrolysis at -0.7 V (see graph legend in Fig 4h). However, we mismarked the potential as -1 V in the figure caption. We are very sorry for our carelessness and have corrected it in the revised manuscript (highlighted with red color in the revised version).*

Comments 3) In Figure 4b, the current densities of samples S4, S5 and S6 increase after a long reaction time, while current densities of other samples decrease with the increase in reaction time. Authors are encouraged to explain this.

Our response: *We thanks a lot for this extremely insightful comment. In fact, as shown in Figure 4b, the current fluctuations of the samples were actually slight during the constant potential electrolysis. However, with the gradual extension of the reaction time, the current density of the samples will inevitably fluctuate with different degrees. The variation trend of current densities between different samples proposed by reviewers is present and also very interesting. Nevertheless, it is not easy to fully understand the trend of the total current density of this reaction. Therefore, we try to analyze the possible reasons of this difference from the following perspectives:*

(1) In our flow-cell system, the gaseous product is in a state of continuous flow, while the liquid product is in the enriched. As the reaction time increases, the accumulation of products inevitably impacts the reaction equilibrium. Consequently, as the reaction proceeds, the activity of generating the liquid product will present a more pronounced decrease compared to that of the gaseous product flowing in the gas chamber.

(2) The results from both DFT calculation and activity experiments indicate that the HER activity of samples S1, S2 and S3 is comparatively lower than that of samples S4, S5 and S6. It suggests that HER serves as a more competitive side reaction for the samples of S4, S5 and S6. Therefore, with an increase in reaction time, the enhancement in HER activity will be more pronounced for samples S4, S5, and S6 compared to the other three samples.

(3) The reduction ability of the adsorbed CO₂ gradually decreases as reaction time prolongs for the six synthesized Bi-Pd IMCs, while the activity of the HER gradually increases. Both factors contribute to the total current density of the reaction. Among these samples, S5 and S6 exhibit higher CO production and fewer liquid products. Consequently, there is no significant decrease in the current density of CO₂RR, but a notable increase in the current density of the strong side reaction HER, resulting in a slight overall increase in total current density. In contrast, for samples S1 and S2, formate is predominantly produced along with minimal gas phase production of CO. As a result, there is a more pronounced decrease in the current density of CO₂RR while weak side reactions HER do not significantly increase. Therefore, there is a slight decrease in total current density for these samples. Additionally, for the samples S3 and S4, there are no noticeable differences between gaseous and liquid product selectivity. Thus, changes in the side reactions HER play key roles in variations observed during electrolysis time prolongation on total current density. Specifically for sample S3, the decline in product activity nearly equals an increase in side reaction HER activity, therefore, no change is observed in the total current density over time during electrolysis. However, in case of sample S4, the increased activity of dominant HER leads to an overall increase in total current density (Figure 4b).

Comments 4) The authors should give infrared data within a wider wavenumber range, and assign other peaks in the infrared spectrum that vary with the potential, such as the variation

of peaks around 1240 cm^{-1} in the infrared spectrum of S6 as shown in Figure 5e.

Our response: We thank for the reviewer's useful comment. According to the reviewer's suggestion, we have provided the FT-infrared full spectra of S1, S4 and S6 in the range of $1000\text{-}4000\text{ cm}^{-1}$ in the revised supplementary information (Fig. S30 in the revised version). As shown in the full spectra below (Figure R5-R7), the prominent peak observed around $3000\text{-}3600\text{ cm}^{-1}$ can be attributed to the stretching vibration peak of water molecules. Additionally, the signals at $2200\text{-}2400\text{ cm}^{-1}$ may be with CO_2 stretching vibrations. To minimize interference from strong water and carbon dioxide signals, our focus lies on comparing signal differences within the range of $1000\text{-}2000\text{ cm}^{-1}$ in the IR spectra, as shown in Figure 5e in the original version (Figure 6e in the revised version). Apart from highlighting key intermediates $^*\text{COOH}$ and $^*\text{OCHO}$ as labelled in the Figure 6e (revised manuscript), notable peaks corresponding to bicarbonate HCO_3^- ($1450, 1650\text{ cm}^{-1}$) [*Adv. Mater.* 2021, 33, 38, 2100143] and carboxylate CO_2^- (1240 cm^{-1}) [*Adv. Energy Mater.* 2021, 11, 41, 2102389] are also observable.

Figure R5 | Potential-resolved in-situ FT-IR full-spectra of Bi_2Pd (S1).

Figure R6 | Potential-resolved in-situ FT-IR full-spectra of Bi_2Pd_5 (S4).

Figure R7 | Potential-resolved in-situ FT-IR full-spectra of BiPd_3 (S6).

Comments 5) The authors used anion exchange membranes to test the performance of CO_2RR , so it is necessary to explain how to avoid the carbon loss caused by the crossover of formate and bicarbonate during the long-term test. A closed carbon mass balance accounting for all carbon should be provided to assure that the results are valid (determination of gaseous CO_2 at the cathode and anode outlet, of organic molecules in the gas phase in the catholyte as well as anolyte, and of carbonate in the catholyte and anolyte necessary).

Our response: We thank for the reviewer's useful comment and totally agree with the reviewer's opinion. Typically, the anion exchange membrane is used when gaseous products dominate CO₂ reduction, while the proton exchange membrane is employed for liquid product formation. In this study, different proportions of liquid and gas products were generated over various Bi-Pd IMCs. However, it should be noted that using a proton exchange membrane may lead to a significant decrease in pH caused by OER on the anode side, which negatively affect CO₂RR activity on the cathode side (Figure R8a~b). This concern was further verified through a comparative experiment with both types of membranes (Figure R8c). Therefore, we utilized anion exchange membrane in this work. At the same time, we addressed concerns raised by reviewers regarding crossover of liquid phase products and carbon loss by providing the following experiments to assure that our results are valid.

We thoroughly analyzed carbon balance from both perspectives of cathode and anode sides. Firstly, we measured the formate content in the catholyte, anolyte and anode-cathode mixture of S1 and S6 after constant potential electrolysis by using ion chromatograph (Figure R9), respectively. As shown in Figure R10, regardless of the amount of produced formate, only trace amounts of formate (4-6% of catholyte) were detected in the anolyte. This indicates that crossover behavior is negligible and can be avoided by measuring the mixed electrolyte. Subsequently, to address the carbon loss phenomenon in the alkaline conditions, carbonate concentrations were determined in both the anode and cathode compartments of S1 and S6. Based on CO₂ input and detected carbonate levels, a rough mass balance accounting for carbon within the reaction system was conducted (Table R1). As expected, there was minimal presence of carbonate in the electrolyte after reaction completion (~0.8-1% carbon loss), suggesting that such losses are likely insignificant. It may be probably owing to the unique three-phase reaction interface in the flow-cell, avoiding the contact loss of CO₂ with the alkaline electrolyte.

Figure R8| a, Schematic of carbonate formation and crossover phenomenon observed using anion exchange membrane (AEM). **b**, Schematics of ion transport and reactions using proton exchange membrane (PEM). **c**, FE at -0.7V (vs. RHE) of the S1 and S6 using AEM and PEM.

Figure R9 | **a**, Calibration curve used for estimation of formate concentration and **b**, ion chromatograms (IC) of formic acid solutions of different concentrations from 0 to 10 mg L⁻¹. **c**, calibration curve used for estimation of formate concentration and **d**, IC of formic acid solutions of different concentrations from 10 to 50 mg L⁻¹. **e**, calibration curve used for estimation of carbonate concentration and **f**, IC of Na₂CO₃ solutions of different concentrations from 0 to 10 mg L⁻¹.

Figure R10 | **a**, IC curves and **b**, the formate concentrations (diluted 100 times) and **c**, carbonate (diluted 40 times) concentrations of anolyte and catholyte of S1 and S6 after the constant potential (-0.7V vs. RHE) electrolysis using AEM. **d**, constant potential (-0.7V vs. RHE) electrolysis of S1 and S6.

Flow Rate of CO ₂	30 mL min ⁻¹
Electrolyte Volume (mixed)	80 mL
Electrolytic Time (t)	2649 s (S1)
	3549 s (S6)
Carbonate Concentration of mixed electrolyte (C)	436.6 mg L ⁻¹ (S1)
	470.3 mg L ⁻¹ (S6)
Carbon Loss Rate $= \frac{C * 80 * 10^{-3} * \frac{44}{60}}{30 * t * \frac{1}{60} * \frac{1}{22.4} * 10^{-3} * 44}$	0.98% (S1)
	0.79% (S6)

Table R1 | Carbon mass balance calculation of S1 and S6.

For reviewer #2:

In this paper, the authors reported preparation of a set of BiPd intermetallic catalysts by solvothermal method, by selecting the amount or the type of Pd salt precursors. The author performed detailed examination for the as prepared catalysts by Aberration corrected TEM and XAFS and DFT calculation and in-situ FTIR were also performed for the catalysts, which is really impressive. There are several reports about the synthesis and application of BiPd intermetallic, which are equally prepared by hydrothermal method, but using different solvent and metal precursors. The concept of the paper, nor the findings seem very new. Thus, the authors may need to highlight the novelty and broad impact of this work.

Comments 1) The reports about BiPd intermetallics are shown below:

([https://doi.org/10.1016/S1872-2067\(21\)63999-2](https://doi.org/10.1016/S1872-2067(21)63999-2); <https://doi.org/10.1002/anie.202109288>; <https://doi.org/10.31635/ccschem.022.202202357>)

Our response: *We thank for the reviewer's useful comment. Bi-based bimetallic catalysts are emerging as fascinating materials with remarkable catalytic properties. Intermetallics provide a desirable platform for atomic-scale structural design and in-depth understanding of the structure-performance correlations in catalyst materials. As perfect catalyst candidates, if a Bi-based intermetallics family with tunable composition and atomic arrangement could be achieved, it is definitely of great significance for a systematic and deeper understanding of the structure-activity correlation for specific catalytic reaction due to their special surface properties and well-defined atomic arrangements. However, a controllable synthesis of a Bi-based intermetallics family is barely investigated because it is a huge challenge to develop a facile method to fine tune the atomic-level surface structure and the compositional stoichiometry of IMCs. In literatures, the intermetallic PdBi nanosheets ([https://doi.org/10.1016/S1872-2067\(21\)63999-2](https://doi.org/10.1016/S1872-2067(21)63999-2)) and Pd₃Bi ordered nanocrystals (<https://doi.org/10.1002/anie.202109288>) have been reported; however, from the point view of synthesis, an extremely complicated multiple-steps template-assisted method was involved for synthesis of PdBi nanosheets, and a general solvothermal process followed by a high temperature annealing treatment (500 °C) was employed for Pd₃Bi intermetallic preparation.*

In respect of catalytic performance, both works focused on catalytic property comparison between fully ordered Pd-Bi products with corresponding disordered ones and gained a similar conclusion that intermetallic products exhibited excellent formate selectivity. The review paper mentioned by this reviewer summarized a synthesis of various Bi-based bimetallic catalysts, which included Pd₃Bi intermetallic and other bimetallic products (e. g. Bi-Sn, Bi-Cu).

Generally speaking, few studies have been implemented to synthesize a family of Bi-Pd IMCs with tunable composition and atomic arrangement via a simple and general wet chemical route because it is difficult to achieve the synthesis of multiple IMCs with different structures under the same synthesis system. Likewise, it also lacks a systematic and deeper understanding of the structure-activity correlation for intermetallic families with different compositions for specific catalytic reaction. Therefore, the development of a simple and general system for the controlled synthesis of ordered IMCs still remains a huge challenge. In this work, we successfully fabricated up to six Bi-Pd ordered IMCs, including Bi₂Pd, BiPd, Bi₃Pd₅, Bi₂Pd₅, Bi₃Pd₈ and BiPd₃ via a EG co-reduction solvothermal method for the first time. A combination of XRD, AC-STEM, XAS and Atomic resolution EDX mapping were used to delicately confirm the structure of Bi-Pd IMCs family. Moreover, using electrocatalytic CO₂ reduction as a model reaction, we systematically investigated the catalytic properties of Bi-Pd IMCs and correlated their particular performances with unique surface structure of each product. The unique full understanding of the specific structure-performance relationship of the IMCs in this work provides a valuable paradigm for precisely modulating the structure of catalyst materials. Therefore, we believe that this work can provide some new insights for the controlled synthesis of IMCs and the selective regulation of catalytic intermediates by different ordered structures. It is the first example for controllable synthesis of six different ordered Bi-Pd IMCs and being studied as a platform to understand the structure-activity relationship, which is highly desirable for catalysis studies. According to the review's suggestion, we further highlight the novelty and broad impact of this work in the revised manuscript as follows.

“Bi-based bimetallic catalysts are emerging as fascinating materials with remarkable catalytic properties. As perfect catalyst candidates, if a Bi-based intermetallics family with tunable composition and atomic arrangement could be achieved, it is definitely of great

significance for a systematic and deeper understanding of the structure-activity correlation for specific catalytic reaction due to their special surface properties and well-defined atomic arrangements. To date, although various IMCs such as Bi-Pb¹⁸, Bi-Mo¹⁹, Bi-Ni²⁰ have been successfully prepared, a controllable synthesis of a Bi-based intermetallics family with tunable composition and atomic arrangement via a simple and general wet chemical route is barely investigated because it is difficult to achieve the synthesis of multiple IMCs with different structures under the same synthesis system. Furthermore, a comprehensive understanding of the structure-activity correlation in catalytic reactions for different phases within intermetallics families remains limited. Therefore, the development of a simple and general system for the controlled synthesis of ordered IMCs still poses a huge challenge.”

Comments 2) Usually, restructuring of the metal catalysts occurs during electrochemical CO₂RR occurs. Thus, the real active site may not be BiPd intermetallics at all. Is that possible that the authors perform in-situ examination of XAS, XRD or perform such characterization for the spent samples.

Our response: *We thank for the reviewer’s useful comment. The restructuring phenomenon usually occurs during electrochemical CO₂RR of metal catalysts which have been widely studied in literatures [J. Am. Chem. Soc. 2023, 145, 18, 10116; ACS Nano 2019, 13, 9, 10818-10825]. Interestingly, most studies focused on unstable catalyst under the electrochemical environment. As a perfect candidate to study the structure-activity correlations in catalytic processes, another overwhelming advantage of intermetallics is their extremely stable properties during electrocatalysis, which have been well demonstrated in lots of studies [Nat. Catal. 2022, 5, 251-258; Adv. Sci. 2020, 7, 1, 1800178]. Although it is not an easy endeavor for us to perform in-situ examination we have indeed considered the possible structural or compositional evolution and performed XRD characterizations of spent Bi-Pd IMCs after electrocatalysis (Figure S18). The results reveal no recognizable structural and compositional changes, suggesting well-maintained crystal structures during the electrolysis. Considering a lack of fine surface information obtained from XRD, we further employed HAADF-STEM to study the surface structure of post-electrocatalyst from an atomic level.*

Taking Bi₂Pd (S1) and BiPd₃ (S6) as examples, which represent the highest formate and CO

conversion, respectively. The HAADF-STEM images were all taken with the electron beam projected along the [001] zone axis, as shown in Figure R11a~b. Both S1 and S6 after reaction keep highly ordered periodic arrangements with alternating bright and dark atoms. By analyzing the special atomic arrangement and the bright contrast files, the ordered arrangements are well matched with the (100) plane of S1 and S6. The further comparison of intensity profiles (Figure R11c~d) measured from HAADF-STEM images along different directions and corresponding fast Fourier-transform (FFT) patterns (Figure R11e~f) are in excellent agreement with the standard crystal spacing. Finally, we simulate the arrangement of (100) plane of S1 and S6 (Figure R11g~h) respectively, which exhibits exactly the same pattern as the HAADF-STEM images we took. All the results indicate that the Bi-Pd IMCs is extremely stable without compositional or structural variations during the long-time electrolysis. We thus believe that real active sites mainly originate from the native Bi-Pd IMCs.

According to the review's suggestion, we have added the following description about the restructuring of the Bi-Pd intermetallics during CO₂RR in the revised manuscript. “Noticeably, the restructuring phenomenon usually occurs during electrochemical CO₂RR of metal catalysts which have been widely studied in literatures [J. Am. Chem. Soc. 2023, 145, 18, 10116; ACS Nano 2019, 13, 9, 10818–10825]. In this work, the XRD patterns of the IMCs before and after long-term stability test manifest no recognizable structure change (Supplementary Fig. S18). We further employ HAADF-STEM to study the surface structure of spent S1 and S6 from an atomic level. As shown in Fig. 5, both S1 and S6 after reaction keep highly ordered periodic arrangements with alternating bright and dark atoms. By analyzing the special atomic arrangement and the bright contrast files (Figs. 5a-b), the ordered arrangements are well matched with the (100) plane of S1 and S6. The further comparison of intensity profiles (Figs. 5c-d) along different directions and corresponding fast Fourier-transform (FFT) patterns (Figs. 5e-f) are in excellent agreement with the standard crystal spacing. Finally, we simulate the arrangement of (100) plane of S1 and S6, respectively, which exhibits exactly the same pattern as the HAADF-STEM images we took (Figs. 5g-h). All the results indicate that the Bi-Pd IMCs is extremely stable without compositional or structural variations during the long-time electrolysis.” (highlighted with red color in the revised manuscript)

Figure R11 | **a-b**, Aberration-corrected HAADF-STEM images of S1 and S6 after reaction. Inset: bright contrast file in the orange box and diagram of atomic arrangement and spacing. **c-d**, Intensity profiles measured from HAADF-STEM images and **e-f**, corresponding FFT pattern and crystal structure (red and green spheres represent Bi and Pd atoms) of S1 and S6 after reaction. **g-h**, Simulated HAADF image of S1 and S6 along [001] directions.

Comments 3) Currently, the production of formate is already not very exciting for electrochemical CO₂RR. The selectivity and current density for C₂ products productions can be up to 80% and 1-2 A cm⁻².

Our response: *We thank for the reviewer's useful comment. Formate product has been widely reported over diverse electrocatalysts in literatures, while they have not been generally demonstrated outside laboratories. The current market price and operating cost have strongly suggested that formate is the most commercially available product. Therefore, there is a*

growing impetus to efficiently convert CO₂ to formate in recent years [J. Am. Chem. Soc. 2021, 143, 14, 5386; Angew. Chem. Int. Ed. 2021, 60, 38, 20627]. In this work, we mainly focus on the synthesis of various Bi-Pd intermetallics via a facile solvothermal method. As ideal catalyst candidates, we then investigate structure-performance relationship over various Bi-Pd intermetallics by employing CO₂ reduction as a model reaction, aiming to uncover the key intermediates that determine the overall reaction activity and selectivity. The results show the surface atomic arrangements endow Bi-Pd IMCs different surface binding affinities and adsorption configuration for various intermediates and eventually different reduction products. The liquid phase product of formate and gas phase product of CO were the dominated product, respectively, while no C₂ products could be able to generate during the electrocatalytic CO₂RR process due to the intrinsic adsorption properties of synthesized six Bi-Pd IMCs. Specifically, Bi₂Pd displays an excellent high formate selectivity. The insights gained from this work not only shed light on the synthesis of intermetallic nanocrystals via a facile solvothermal method but also provide an important knowledge framework that guides the rational design of architecturally sophisticated multimetallic nanostructures toward optimization of catalytic molecular transformations during heterogenous catalysis.

Comments 4) However, it remains a huge challenge in the controllable synthesis of IMCs, as diffusing a metal atom into a lattice of another metal to form a regular metal occupancy is no picnic. The formation mechanism for the intermetallic is not well supported by data. Can the authors monitor the formation dynamics for such intermetallics? Why do you have to change the type of Pd precursors to regulate the ratio of Pd to Bi in the intermetallics, but just using only one single Pd precursors.

Our response: We thank for the reviewer's extremely insightful comments. In the synthesis of Bi-Pd IMCs, we have run hundreds of experiments to explore the controllable synthesis strategy of ordered Bi-Pd IMCs with different atomic compositions and arrangements by changing the reaction parameters including reaction time, temperature, Bi/Pd molar ratio, and the type of Pd precursors. It is not an easy endeavor to monitor the formation dynamics for such intermetallics, which requires advanced techniques such as operando XAFS and in situ HAADF-STEM during solvothermal process. According to the reviewer's suggestions, we

tried to monitor and analyze the growth process of Bi-Pd IMCs by powder X-ray diffraction.

We collected the products obtained at different reaction times (1, 2, 3, 4, 5, 6, 7, 8, 9 and 12 h). As shown in Figure R12-14, time-dependent XRD patterns of various intermetallics could be roughly divided into three different stages: initial nucleation, metallic interdiffusion (phase transition), intraparticle atomic diffusion (ripening) (highlighted in different background color). For Bi_2Pd , amorphous substance was obtained at 1h, which then evolved into a mixture of BiPd and Bi_2Pd after 2 h reaction. With the prolonging of the reaction time, more Bi atoms diffused into BiPd lattice and a Bi_2Pd intermetallic formed after 4 h reaction. The Bi_2Pd intermetallic further underwent an intraparticle atomic diffusion and crystallization process to transform into final product (S1). Interestingly, monoclinic phase and tetragonal phase demonstrated a reversible phase transformation, which always co-existed even further prolonging the reaction time, probably due a comparable thermal dynamic stability. For Bi_2Pd_5 synthesis, beside Pd formation, a simultaneous generation of BiOCl product was also observed within the initial 5 h reaction due to the introduction of chlorine-containing Pd precursor (sodium tetrachloropalladate, Na_2PdCl_4). As the reaction proceeded, BiOCl was sequentially reduced into Bi, which then experienced atomic interdiffusion with Pd under solvothermal alloying conditions, leading to the formation of BiPd₃ intermetallic after 6 h reaction. As more Bi generation and diffusion, BiPd₃ gradually converted into stable Bi_2Pd_5 intermetallic product after 8 h reaction. Slightly different with Bi_2Pd and Bi_2Pd_5 , the formation of BiPd₃ looks more straightforward. Both Pd and BiPd₃ phase appeared in the first 1 h reaction, which then underwent a further Bi deposition and atomic diffusion to generate a stable BiPd₃.

In addition, we also studied the influence of Pd precursor. Besides Na_2PdCl_4 , $\text{Pd}(\text{NO}_3)_2$, $\text{Pd}(\text{acac})_2$, palladium(II) acetate ($\text{Pd}(\text{acetate})_2$) and sodium tetrachloropalladate (Na_2PdCl_4) were also used as Pd precursors to grow Bi-Pd intermetallic keep other reaction conditions identical. Taking Bi_2Pd_5 as an example, time-dependent XRD patterns (Figure R15-17) show it failed to get Bi_2Pd_5 products by switching Na_2PdCl_4 to another three Pd precursors, which may be explained by their solubility in reaction solvent. For example, when more soluble $\text{Pd}(\text{NO}_3)_2$ is used, the release rate of Pd^{2+} is fast, resulting in a quick nucleation of BiPd₃/and Pd with 1h. However, when a weak solubility palladium acetate was used, the release rate

became slow, and only Pd and some organic impurities appeared at 1h. In turn, when indissoluble palladium acetylacetonate is used, the nucleation is slower. No reduction product was formed in the first 1 h. A very limited solubility of Pd(acac)₂ also restricted the reduction rate of Pd(II) ions, which extended the growth period of Pd to 12 h.

Due to the limited space of this manuscript, we do not include a detailed discussion of those data of the formation mechanism for the intermetallic in the main body. According to the reviewer's suggestion, we have added relevant discussions in our revised manuscript as following, and attached the data in the revised supplementary information (Figs. S1-S6 in the revised version). "In the synthesis of Bi-Pd IMCs, we have run hundreds of experiments to explore the controllable synthesis strategy of ordered Bi-Pd IMCs with different atomic compositions and arrangements by changing the reaction parameters including reaction time, temperature, Bi/Pd molar ratio, and the type of Pd precursors (Supplementary Figs. S1-S6). Six different Bi-Pd ordered intermetallic compounds (IMCs) have been successfully prepared by co-reduction of Bi and Pd salts via a general and facile solvothermal method (S1~S6)."

Figure R12 | XRD patterns of the S1-Bi₂Pd products obtained at various reaction times.

Figure R13 | XRD patterns of the S4-Bi₂Pd₅ products obtained at various reaction times.

Figure R14 | XRD patterns of the S6-BiPd₃ products obtained at various reaction times.

Figure R15 | XRD patterns of the S4-Bi₂Pd₅ products using Pd(OAc)₂ (Pd(acetate)₂) as the Pd precursor obtained at various reaction times.

Figure R16 | XRD patterns of the S4-Bi₂Pd₅ products using Pd(NO₃)₂·2H₂O as the Pd precursor obtained at various reaction times.

Figure R17 | XRD patterns of the S4-Bi₂Pd₅ products using C₁₀H₁₄O₄Pd (Pd(acac)₂) as the Pd precursor obtained at various reaction times.

Comments 5) The clarity of the figures provides is quite low and need to be improved.

Our response: We thank for the reviewer's useful comment. According to the suggestion of the reviewer, the clarity of all the figures provided in the manuscript has been improved and we have replaced all the figures in the original version as requested. (highlighted in the red color in the revised version)

For reviewer #3:

The DFT calculation section of the article provides detailed research on HER and CO₂RR reactions occurring on S1, S4, and S6, but there are still several issues:

Comments 1) The content of Supplementary Fig. S18 at line 236 does not correspond to the HER reaction described in the text.

Our response: *We thank for the reviewer's preciseness. We are very sorry for our misspelling of graphic Numbers. "Fig. S18" should be Fig. S16b in the original manuscript (line 235-236)", which has been changed to "Fig. S25b" in the revised manuscript due to the added supplementary figures. (highlighted in the red color in the revised version)*

Comments 2) At lines 246 and 247, it is stated that "the next hydrogenation to form * OCHO and * COOH are rate determining steps for the formation of form and CO, possessing the highest energy barrier". However, the corresponding Supplementary Figures S17 and S18 only indicate the transition state energy barrier but do not indicate the thermodynamic energy barrier for this step of the reaction. Moreover, in Figure S17a, it appears that the energy barrier for * OCHO to * HCOOH is actually higher, which is inconsistent with the textual description; And there is a syntax error in "While the different absorption configurations change the subsequence reaction paths for separate products."

Our response: *We thank for the reviewer's extremely valuable comment. Generally, it is proposed the electron transfer to adsorbed CO₂ (*CO₂) and following hydrogenation for *OCHO and *COOH generation are the rate-determining steps for formate and CO product, respectively, which has well demonstrated by previous studies. [Chem. Sci., 2017, 8: 1090-1096; Adv. Mater., 2021, 33: 2100143.] In this work, we also tried to use Tafel analysis to investigate the kinetics or mechanistic insight over catalysts, as seen in Fig R18, while almost all samples possessed an extremely large Tafel slope, also indicating the electron transfer to CO₂ and the next hydrogenation probably was the rate-determining step during CO₂RR. To well understand the structure-dependent selectivity of different IMCs, we employed DFT calculations to simulate three reaction pathways over typical Bi₂Pd (S1), Bi₂Pd₅ (S4), and BiPd₃ (S6) based on their specific reduction products (H₂, CO, formate). As shown in Figs.*

S16-S18 in the original version (Figs. S25~S27 in the revised version), we demonstrated the energy barrier for transition state (ΔTS) during different reaction pathways, which had no direct relationship with the thermodynamic energy barrier. We are sorry for our imprecise description to make the reviewer confuse. In order to make it more clearly, we further simulate the thermodynamic energy barriers for the first electron-transfer step ($\Delta G1$) and the second electron-transfer step ($\Delta G2$) in the CO and HCOOH generation paths over S1, S4 and S6 samples. The results were summarized in following Table R2. The results demonstrate that both ΔTS and $\Delta G1$ are higher than $\Delta G2$, which indicate that the first electron-transfer step is still the rate determining step of the reaction without considering the transition state. However, as mentioned by the reviewer, there is a difference for the HCOOH pathway in S1 where $\Delta G2$ (1.59 eV) exhibits a slightly larger energy barrier compared to $\Delta G1$ (1.28 eV), and a higher energy barrier for *OCHO to *HCOOH was also observed. However, for HCOOH generation, S1 exhibited the most favorable pathway compared with another two samples. Actually, the real electrochemical process is extremely complex, and a comprehensive DFT calculations requires consideration of various experimental details such as specific alkaline electrolyte environments. During the simulation, we have considered hydrolyzation-protonation processes and found that it makes the protonation of the *OCHO intermediates to the *HCOOH more difficult under alkaline conditions; however, they are more likely to evolve into formate products in the actual reaction, which might can explain why generating *HCOOH from *OCHO intermediates needs a higher energy in our highly alkaline environments due to the favorable formation of formate species rather than HCOOH in most cases. Regarding higher reaction energies observed here compared to those predicted by hydrogen electrode models, it can be attributed to our adoption of an E-R reaction mechanism [Nat. Commun., 2020, 11:2256], which is different from hydrogen electrode models involving proton-hydrogen-electron pairs during reactions.

Finally, we are very sorry for the grammatical errors in “While the different absorption configurations change the subsequence reaction paths for separate products” in the original manuscript (line 247-248). We have corrected it to “Different absorption configurations change the subsequence reaction paths for separate products” in the revised manuscript. (highlighted in the red color in the revised version)

Figure R18 | Tafel plots of Bi-Pd IMCs in flow cell.

CO ₂ →CO	ΔTS (ev)	ΔG1 (ev)	ΔG2 (ev)
S1	1.73	0.93	-0.22
S4	1.20	1.27	-0.40
S6	1.14	0.96	-0.89
CO ₂ →HCOOH	ΔTS (ev)	ΔG1 (ev)	ΔG2 (ev)
S1	1.45	1.28	1.59
S4	3.57	1.31	1.12
S6	2.09	1.56	1.28

Table R2 | DFT-calculated free energy barrier of three elementary step for S1, S4 and S6 in CO₂RR.

Comments 3) At line 441, Did you increase kpoints when calculating DOS? Suggest explanation for this.

Our response: *We thank for the reviewer's useful comment. In the catalytic process, our focus lies predominantly on the electronic state of the material and the shift in the D-band center of the active atom. However, we keep K-points consistent for catalytic process and DOS calculations by using the same number of K-points. This is primarily due to the utilization of a cut-off surface rather than its unit structure form in DOS calculations, resulting in a significantly larger system and an exponential increase in computational requirements with additional K-points. For our current system, K-points of 2*2*1 and 3*3*1 adequately suffice for accurate DOS calculations.*

The revised version has already been submitted online.

Thank you for your consideration!

REVIEWER COMMENTS

Reviewer #1 (Remarks to the Author):

Authors prepared intermetallic BiPd materials and evaluated them as electrocatalysts for electrochemical CO₂ reduction to formate. Although the materials were well characterized, the same concept has been reported in *Angew. Chem., Int. Ed.* (2021, 60, 21741-21745), where both intermetallic and solid–solution BiPd were prepared and demonstrated to efficiently convert CO₂ into formate. Thus there is a big concern regarding the novelty.

Moreover, formate is a simple 2 electron transferred product during CO₂ reduction and has been widely and efficiently generated over cheap tin, indium, and bismuth metals. It is unnecessary to add noble Pd, which largely increases the CO₂RR catalyst cost and prevents them from practical application.

In the response letter, authors claimed that the anion exchange membrane is used when gaseous products dominate CO₂ reduction, while the proton exchange membrane is employed for liquid product formation. This statement is not true and baseless. In fact, anion exchange membrane rather than proton exchange membrane is widely employed in CO₂ electrolyzers.

As for formate crossover from cathode to anode, authors claimed a small amounts of formate (4-6% of catholyte) and a small carbon loss (~0.8-1%) but with a rather small current densities of below 30 mA cm⁻² (compared to over 200 mA cm⁻² required for industrial CO₂ electrolysis) and within less than one hour electrolysis. How about over 100 hour electrolysis at an industrial relevant current density? Serious formate crossover and carbon loss will be very obvious in the current flow cell setup

Therefore I can not recommend this study for publication in Nature Communications.

Reviewer #2 (Remarks to the Author):

Accept as is.

Reviewer #3 (Remarks to the Author):

The author has answered my question properly and I recommend publishing it on Nature Communications.

Dear reviewers,

We would like to thank you for the prompt and comprehensive review of our manuscript. These insightful comments have greatly contributed to the improvement of our work. We have carefully studied the comments, supplemented additional experiments and provided a substantive discussion on the obtained results. Please find our detailed responses to each comment from the reviewer below, along with corresponding revisions highlighted in *red* text in the revised manuscript.

For reviewer #1:

Comments 1) Authors prepared intermetallic BiPd materials and evaluated them as electrocatalysts for electrochemical CO₂ reduction to formate. Although the materials were well characterized, the same concept has been reported in *Angew. Chem., Int. Ed.* (2021, 60, 21741-21745), where both intermetallic and solid–solution BiPd were prepared and demonstrated to efficiently convert CO₂ into formate. Thus, there is a big concern regarding the novelty.

Our response: *We appreciate the reviewer’s valuable comment. We acknowledge that some Bi-Pd intermetallics have been already investigated, and we cited the relevant publications on this topic in the manuscript. At first glance, it might be a straightforward conclusion to draw that our work is a simple extension. However, the key novelty of this work lies in precious control of BiPd intermetallics with various stoichiometries almost covering the whole range of Bi-Pd phase diagram through a general solvothermal method that will broadly excite the nanoscience community, which remained unexplored in the previous work. Furthermore, our work offers insights into unraveling detailed structure-activity correlations, which should be considered as a significant new progress in this area as well. We firmly believe that this work represents an unprecedented level of precision and versatility in achieving the structural and compositional control over BiPd nanocatalysts at an atomic level and provides valuable insightful knowledge that guides the rational optimization of electrocatalysis for fuel cell applications. Although reaching a consensus on whether this work possesses sufficient novelty for publication in *Nature Communications* may be challenging, we would like to summarize the relevant literatures on Bi-Pd IMCs and make a statement as follows¹⁻⁷.*

(1) Most literatures only achieved the fabrication of Bi-Pd intermetallic compounds (IMCs)

with no more than two ordered states. There is a lack of research on synthesizing a family of Bi-Pd IMCs with tunable composition and atomic arrangement. The controlled synthesis of intermetallic compounds has long been a challenging task, particularly at relatively low temperatures using simple wet chemical routes. Therefore, it is both innovative and instructive to develop a co-reduction system capable of producing up to six different Bi-Pd IMCs, some of which have not been reported previously. Additionally, the direct characterization and determination of the ordered structure of Bi-Pd IMCs at the atomic level also remains a big challenge. There are limited studies that can consistently identify accurate structures based on crystal structure analysis, atomic arrangement investigation, elemental analysis and electronic structure examination.

(2) The reported structure-activity correlations in this topic concentrated on activity difference between intermetallic compounds and their corresponding disordered alloys in terms of their degree of order. For instance, the phase-dependent electrocatalytic CO₂ reduction on Pd₃Bi nanocrystals mentioned by the reviewer [Li et al., *Angew. Chem., Int. Ed.*, 2021, 60, 2174-21745]. Highly ordered PtFe intermetallics exhibits enhanced electrocatalytic properties and the ordering degree-dependent performance can be ascribed to the compressive strain effect induced by the intermetallic PtFe core with smaller lattice parameters, and the more thermodynamically stable intermetallic structure compared to disordered alloys [Liang et al., *Nature Communications*, 2022, 13, 7654; Liang et al., *Small*, 2022, 2202916]. There is a lack of systematic and comprehensive studies to explore the effects of different ordered interfaces at the atomic level. Therefore, it is imperative to synthesize a diverse range of intermetallic compounds with varying atomic arrangements in order to investigate their reaction mechanisms at an atomic level.

(3) Although there have been reports on electrocatalytic reduction of CO₂ to formate by certain Bi-Pd intermetallic compounds (IMCs), the selectivity mechanism of these specific structures, as well as the activity and selectivity of other Bi-Pd IMCs with different arrangements and proportions, remain unexplored.

Different with reported studies, this work presents a general polyol-reduction system for tailoring the atomic arrangement and compositional stoichiometry of six different Bi-Pd intermetallic compounds, including Bi₂Pd, BiPd, Bi₃Pd₅, Bi₂Pd₅, Bi₃Pd₈ and BiPd₃. Our

successful synthesis and well characterization of a series of Bi-Pd IMCs reached an unprecedented level for the precise control of intermetallic alloy structures in nano community. This unique platform enables an in-depth exploration of the structure-selectivity relationships. Furthermore, employing electrocatalytic CO₂RR as a model reaction, we systematically elucidate the distinct selectivity of C1 products on six Bi-Pd IMCs and establish correlations between adsorption configuration of key intermediates and their corresponding atomic arrangement. The profound understanding gained from this work regarding the specific structure-performance relationship of IMCs offers valuable insights into precise modulation of catalyst materials structures. Therefore, we believe that our findings provide novel perspectives on controlled synthesis of IMCs and selective regulation of catalytic intermediates through different ordered structures. Notably, this is the first instance demonstrating controllable synthesis of six distinct ordered Bi-Pd IMCs while investigating their structure-activity relationship; thus, representing an innovative and instructive contribution to current research endeavors.

Comments 2) Moreover, formate is a simple 2 electron transferred product during CO₂ reduction and has been widely and efficiently generated over cheap tin, indium, and bismuth metals. It is unnecessary to add noble Pd, which largely increases the CO₂RR catalyst cost and prevents them from practical application.

Our response: We appreciate for the reviewer's concerns and totally agree with the reviewer's opinion that efficient conversion of CO₂ to formate has been achieved over certain metals, such as Sn, In, and Bi⁸⁻¹⁴. If only the market price and practical application are considered, there is really no necessity to introduce the precious metal palladium into this system. However, like our response for comment 1, this work essentially aimed at developing fundamental understanding of intriguing structural control and profound understanding their specific structure-performance relationship of BiPd intermetallics. Achieving record-breaking catalytic performances is not the goal of this work and we did not intend to claim that Pd introduction beats the state-of-art catalysts. Therefore, taking CO₂ reduction reaction (CO₂RR) as a model reaction, we systematically investigated and compared the catalytic properties of six Bi-Pd IMCs, revealing distinct selectivity towards C1 products on these samples. While hexagonal BiPd exhibits high selectivity for formate (FE>80 %), orthorhombic BiPd₃

demonstrates a pronounced preference for CO ($FE \approx 70\%$) and monoclinic Bi_2Pd_5 favors higher selectivity for H_2 ($FE \approx 50\%$) on gas diffusion electrode (GDE). Subsequently, we elucidate the structure-activity relationship between adsorption configuration of key intermediates and their corresponding atomic arrangement. As fundamental research, our work primarily focuses on precise atomic regulation and comprehensive understanding of structure-activity relationships in ordered intermetallic compounds—an area that still requires significant advancements before industrial applications can be realized. The precise control of atomic structures of different ordered Bi-Pd IMCs also would be potentially well used in many other fields like ORR and NRR.

Comments 3) In the response letter, authors claimed that the anion exchange membrane is used when gaseous products dominate CO_2 reduction, while the proton exchange membrane is employed for liquid product formation. This statement is not true and baseless. In fact, anion exchange membrane rather than proton exchange membrane is widely employed in CO_2 electrolyzers. As for formate crossover from cathode to anode, authors claimed a small amount of formate (4-6% of catholyte) and a small carbon loss ($\sim 0.8-1\%$) but with a rather small current densities of below 30 mA cm^{-2} (compared to over 200 mA cm^{-2} required for industrial CO_2 electrolysis) and within less than one hour electrolysis. How about over 100 hours electrolysis at an industrial relevant current density? Serious formate crossover and carbon loss will be very obvious in the current flow cell setup.

Our response: *We greatly appreciate for the reviewer's insightful comment. The ion exchange membrane plays a key role in cell performance, and we totally agree with reviewer's opinion that the anion exchange membranes are currently widely employed under the most neutral or alkaline conditions. We sincerely apologize for our previously hasty and inadequate conclusion, which was based on some of literature evidence. For examples, proton exchange membrane is employed for formate production over Pb1Cu-SAAs [Zeng et al., Nature Nanotechnology, 2021, 16, 1386-1393] and 2D-Bi catalyst in flow cell under neutral or alkaline conditions. [Wang et al., Nature Energy, 2019, 4, 776-785]. While anion exchange membrane is used for CO production over 2H/fcc-Heterophase AuCu Nanostructures [Zhang et al., Advanced Materials, 2023, 35, 2304414]. We also fully agree with reviewer's viewpoint that the utilization of anion exchange membranes is inevitably accompanied by product*

crossover or carbon loss. The migration of liquid products through the membrane, driven by electroosmotic drag or diffusion due to concentration gradients, can result in their dilution in the anodic stream, thereby increasing the separation cost, or their oxidation back to CO₂, leading to decreased device efficiency¹⁵. To date, there are numerous studies specifically addressing these issues using a solid electrolyte reactor, strong acid electrolyte or employing bipolar membranes [Wang et al., Nature Catalysis, 2022, 5, 288-299; Edward H. Sargent et al., Science, 2021, 372, 1074-1078; David Sinton et al., Joule, 2022, 6, 1333-1343]. Despite these advancements have been made in CO₂ electroreduction¹⁶⁻²⁰ and resulting in highly concentrated liquid products collected in the cathode chamber, the challenge of product crossover persists.

Therefore, the claim of “a small amount of formate crossover (4-6 % of catholyte) and carbon loss (~0.8-1 %) below 30 mA cm⁻² within one hour at -0.7V (vs RHE)” was indeed not rigorous at this stage. According to the reviewer's comment, we conducted a long-term constant current electrolysis at high current densities (100 mA cm⁻²) (Figure R1). The results clearly demonstrate a rapid decrease in formate Faradaic efficiencies and an increase in hydrogen production over 4 hours, accompanied by significant product crossover (~16 %) and carbon loss (~11 %), which have substantial implications for cell performance and cannot be overlooked any longer. This decline in performance is primarily attributed to electrode flooding, product crossover, carbon loss and salt accumulations resulting from physicochemical changes of the membrane and the gas diffusion layer during prolonged electrolysis at high current densities in alkaline media. It is worth noting that this product crossover and carbon loss will become more serious with the reaction time proceeding, if no treatment is carried out timely.

Figure R1 | Constant current (-100 mA cm^{-2}) electrolysis in flow cell.

Thus, we have tried several strategies, including regular electrolyte refreshing, membrane replacement, and retreatment of the gas diffusion layer to mitigate this performance deterioration. Once a noticeable decrease in activity was observed after a certain period of electrolysis, we replaced the utilized electrolyte and membrane, as well as elaborately washing the cathode surface of the gas diffusion layer. After nearly a month of experimentation, as anticipated, this simple treatment resulted in a substantial reduction in operating voltage and an obvious recovery in formate Faradaic efficiencies (FEs), as depicted in Figure R2a. Furthermore, through periodic treatments, the system exhibited superior stability for nearly 60 hours with small changes in formate crossover ($\sim 13\%$) and carbon loss ($\sim 9\%$), and a well-maintained formate selectivity ($FE \sim 70\%$). These findings indicate that these treatments can effectively alleviate the product crossover and carbon loss to some extent during alkaline CO_2RR . However, prolonged usage leads to gradual hydrophobicity loss of the gas diffusion layer, as evidenced by the decreasing contact angle (Figure R2b). Consequently, this hinders CO_2 diffusion to reaction sites while promoting hydrogen evolution reaction and the electrode flooding at the reactive three-phase interface, thus limiting the further stability testing within the current flow cell setup.

Figure R2 | **a**, Long-term stability test of S1 at -100 mA cm^{-2} in flow cell. **b**, Contact angles of the cathode electrode surface at different reaction time and after treatment.

For reviewer #2:

Comments: Accept as is.

Our response: We greatly appreciate for the reviewer's positive comment.

For reviewer #3:

Comments: The author has answered my question properly and I recommend publishing it on *Nature Communications*.

Our response: We greatly appreciate for the reviewer's positive comment.

References

1. Jia, L., *et al.* Phase-dependent electrocatalytic CO₂ reduction on Pd₃Bi Nanocrystals. *Angew. Chem. Int. Ed.* **60**, 21741-21745 (2021).
2. Zhou, M., *et al.* Improvement of oxygen reduction performance in alkaline media by tuning phase structure of Pd-Bi nanocatalysts. *J. Am. Chem. Soc.* **143**, 15891-15897 (2021).
3. Wang, X., *et al.* Nanoporous intermetallic Pd₃Bi for efficient electrochemical nitrogen reduction. *Adv. Mater.* **33**, 2007733-2007740 (2021).

4. Shen, T., *et al.* Tailoring the antipoisoning performance of Pd for formic acid electrooxidation via an ordered PdBi intermetallic. *ACS Catal.* **10**, 9977-9985 (2020).
5. Sun, D., *et al.* Ordered intermetallic Pd₃Bi prepared by an electrochemically induced phase transformation for oxygen reduction electrocatalysis. *ACS Nano* **13**, 10818-10825 (2019).
6. Wang, Y. & Hall, A. Pulsed electrodeposition of metastable Pd₃₁Bi₁₂ nanoparticles for oxygen reduction electrocatalysis. *ACS Energy Lett.* **5**, 17-22 (2019).
7. Xie, L., *et al.* Regulating Pd-catalysis for electrocatalytic CO₂ reduction to formate via intermetallic PdBi nanosheets. *Chinese J. Catal.* **43**, 1680-1686 (2022).
8. Bi, J., *et al.* High-Rate CO₂ electrolysis to formic acid over a wide potential window: an electrocatalyst comprised of indium nanoparticles on chitosan-derived graphene. *Angew. Chem. Int. Ed.* **62**, e202307612 (2023).
9. Jia, B., *et al.* Indium cyanamide for industrial-grade CO₂ electroreduction to formic acid. *J. Am. Chem. Soc.* **145**, 14101-14111 (2023).
10. Jiang, Z., *et al.* A bismuth-based zeolitic organic framework with coordination-linked metal cages for efficient electrocatalytic CO₂ reduction to HCOOH. *Angew. Chem. Int. Ed.* **62**, e202311223 (2023).
11. Lin, L., *et al.* A nanocomposite of bismuth clusters and Bi₂O₂CO₃ sheets for highly efficient electrocatalytic reduction of CO₂ to formate. *Angew. Chem. Int. Ed.* **62**, e202214959 (2023).
12. Lv, L., *et al.* Coordinating the edge defects of bismuth with sulfur for enhanced CO₂ electroreduction to formate. *Angew. Chem. Int. Ed.* **62**, e202303117 (2023).
13. Xue, H., Zhao, Z.H., Liao, P.Q. & Chen, X.M. "Ship-in-a-Bottle" integration of ditiin(IV) sites into a metal-organic framework for boosting electroreduction of CO₂ in acidic electrolyte. *J. Am. Chem. Soc.* **145**, 16978-16982 (2023).
14. Zhu, J., *et al.* Surface passivation for highly active, selective, stable, and scalable CO₂ electroreduction. *Nat. Commun.* **14**, 4670 (2023).
15. Hasa, B., *et al.* Benchmarking anion-exchange membranes for electrocatalytic carbon monoxide reduction. *Chem Catal.* **3**, 100450-100464 (2023).
16. Xu, Y., *et al.* A microchanneled solid electrolyte for carbon-efficient CO₂ electrolysis. *Joule* **6**, 1333-1343 (2022).
17. Tesler, A.B., *et al.* Long-term stability of aerophilic metallic surfaces underwater. *Nat. Mater.* **22**, 1548-1555 (2023).
18. Kim, J.Y.T., *et al.* Recovering carbon losses in CO₂ electrolysis using a solid electrolyte reactor. *Nat. Catal.* **5**, 288-299 (2022).

19. Fan, M., *et al.* Single-site decorated copper enables energy-and carbon-efficient CO₂ methanation in acidic conditions. *Nat. Commun.* **14**, 3314 (2023).
20. Cui, L., *et al.* An anti-electrowetting carbon film electrode with self-sustained aeration for industrial H₂O₂ electrosynthesis. *Energy Environ. Sci.*, DOI <https://doi.org/10.1039/D1033EE03223J> (2024).

The revised version has already been submitted online.

Thank you for your consideration!